# DEMYSTIFYING HOW SELF-SUPERVISED FEATURES IMPROVE TRAINING FROM NOISY LABELS

## ABSTRACT

The advancement of self-supervised learning (SSL) motivates researchers to apply SSL on other tasks such as learning with noisy labels. Recent literature indicates that methods built on SSL features can substantially improve the performance of learning with noisy labels. Nonetheless, the deeper reasons why (and how) SSL features benefit the training from noisy labels are less understood. In this paper, we study why and how self-supervised features help networks resist label noise using both theoretical analyses and numerical experiments. Our result shows that, given a quality encoder pre-trained from SSL, a simple linear layer trained by the cross-entropy loss is theoretically robust to symmetric label noise. Further, we provide insights for how knowledge distilled from SSL features can alleviate the over-fitting problem. We hope our work provides a better understanding for learning with noisy labels from the perspective of self-supervised learning and can potentially serve as a guideline for further research.

## 1 INTRODUCTION

Deep Neural Networks (DNNs) have achieved remarkable performance in many areas including speech recognition (Graves et al., 2013), computer vision (Krizhevsky et al., 2012; Lotter et al., 2016), natural language processing (Zhang & LeCun, 2015) *etc*. The high-achieving performance often builds on the availability of quality-annotated datasets. In real world scenario, data annotation inevitably brings in label noise which degrades the performance of the network, primarily due to DNNs' capability in "memorizing" noisy labels (Zhang et al., 2016).

In the past few years, a number of methods have been proposed to tackle the problem of learning with label noise including robust loss design (Ghosh et al., 2017; Zhang & Sabuncu, 2018; Liu & Guo, 2020), sample selection (Han et al., 2018; Yu et al., 2019; Cheng et al., 2021) and noise transition matrix estimation (Patrini et al., 2017; Zhu et al., 2021b). Among all these methods, arguably the most efficient treatment is to adopt robust losses, since sample selection and noise transition matrix estimation always involve training multiple networks or need multi-stage training. Nonetheless, though the designed losses are theoretically proven robust, they often suffer from significant performance drop when noise rate is high (Wang et al., 2019; Ma et al., 2020; Cheng et al., 2021; Zhu et al., 2021a), hinting that the ability of converging to the optimal classifier is also important.

Very recent works (Zheltonozhskii et al., 2021; Nodet et al., 2021; Ghosh & Lan, 2021; Yao et al., 2021; Tan et al., 2021) started applying self-supervised learning to solving the problem of learning from noisy labels. The experiments show that methods built on the self-supervised features can achieve exceptional performance even when the noise rate is high and largely outperform previously reported SOTA approaches. Despite the empirical observations, the reasons why self-supervised features lead to significant performance improvement are not well understood. In this paper, we provide theoretical insights to understand how self-supervised features improve classification with label noise and perform extensive experiments to support our theory. Our analysis provides a new understanding on learning with noisy labels from the perspective of self-supervised learning. We summarize our main contributions below:

- We theoretically and experimentally show that by using self-supervised features to fine-tune the network on noisy datasets, Cross Entropy itself is robust to label noise (Theorem 1–2). The theory also answers the question of whether or not to fix the encoder when performing fine-tuning.

- We theoretically and experimentally show that by using self-supervised features, a regularizer commonly used in knowledge distillation (Hinton et al., 2015) (where the dataset does not contain label noise) can greatly alleviate over-fitting problem of DNN on noisy datasets (Theorem 3–4).

## 1.1 RELATED WORKS

**Learning with Noisy Labels:** Due to the over-fitting problem of DNN, many works design robust loss to improve the robustness of neural networks. (Ghosh et al., 2017) proves MAE is inherently robust to label noise. However, MAE has a severe under-fitting problem. (Zhang & Sabuncu, 2018) propose a loss which can combine both the advantage of MAE and CE, exhibiting good performance on noisy datasets. (Liu & Guo, 2020) introduces peer loss, which is proven statistically robust to label noise without knowing noise rate. The extension of peer loss also shows good performance on instance-dependent label noise (Cheng et al., 2021; Zhu et al., 2021a). Another efficient approach to combat label noise is by sample selection (Jiang et al., 2018; Han et al., 2018; Yu et al., 2019; Northcutt et al., 2021; Yao et al., 2020; Wei et al., 2020; Zhang et al., 2020). These methods regard "small loss" examples as clean ones and always involve training multiple networks to select clean samples. Semi-supervised learning is also popular and effective on learning with noisy labels in recent years. Some works (Li et al., 2020; Nguyen et al., 2020) first perform clustering on the sample loss and divide the samples into clean ones and noisy ones. Then drop the labels of the "noisy samples" and perform semi-supervised learning on all the samples.

**Self-Supervised Learning:** The goal of self-supervised learning (SSL) is to learn good presentation without using the information of the labels. Generally, the methods of SSL can be divided into two categories: designing pretext tasks or designing loss functions. The designed tasks or losses do not involve any labels. Some popular tasks include patch orderings (Doersch et al., 2015; Noroozi & Favaro, 2016), tracking (Wang & Gupta, 2015) or clustering features (Caron et al., 2018; 2019). However, the SSL performance of pretext tasks is limited. Recent SOTA methods for SSL is by designing contrastive loss functions. The representative works include Moco (He et al., 2020) and SimCLR (Chen et al., 2020) which train neural networks based on InfoNCE loss (Oord et al., 2018). In our paper, we also adopt InfoNCE for performing self-supervised training to get SSL pre-trained features. The first part of our paper relates to the works that apply SSL features to perform fine-tuning on noisy dataset (Nodet et al., 2021; Ghosh & Lan, 2021) and our goal is to build theoretical understanding on this aspect.

**Knowledge Distillation:** The second part of our paper is very related to the research field of knowledge distillation (KD). The original idea of KD can be traced back to model compression (Buciluǎ et al., 2006), where authors demonstrate the knowledge acquired by a large ensemble of models can be transferred to a single small model. (Hinton et al., 2015) generalize this idea to neural networks and show a small, shallow network can be improved through a teacher-student framework. Due to its great applicability, KD has gained more and more attention in recent years and numerous methods have been proposed to perform efficient distillation (Mirzadeh et al., 2020; Zhang et al., 2018; 2019). However, the dataset used in KD is assumed to be clean. Thus it is hard to connect KD with learning with noisy labels. In this paper, we theoretically and experimentally show that a regularizer generally used in KD (Park et al., 2019) can alleviate the over-fitting problem on noisy data by using SSL features which offers a new alternative for dealing with label noise.

## 2 PRELIMINARY

We introduce preliminaries and notations including definitions and problem formulation.

**Problem Formulation:** Consider a classification problem on a set of $N$ training examples denoted by $D := \{(x_n, y_n)\}_{n \in [N]}$, where $[N] := \{1, 2, \cdots, N\}$ is the set of example indices. Examples $(x_n, y_n)$ are drawn according to random variables $(X, Y)$ from a joint distribution $\mathcal{D}$. The classification task aims to identify a classifier $C$ that maps $X$ to $Y$ accurately. Our theoretical analyses focus on binary classifications thus $Y \in \{0, 1\}$. In real-world applications, the learner can only observe noisy labels. For instance, human annotators may wrongly label some images containing cats as ones that contain dogs accidentally or irresponsibly. The label noise of each instance is assumed to be class-dependent (Liu & Tao, 2015), i.e., $\mathbb{P}(\widetilde{Y}|Y) = \mathbb{P}(\widetilde{Y}|X, Y)$. Thus the error rates are defined as $e_+ = \mathbb{P}(\widetilde{Y} = 0|Y = 1), e_- = \mathbb{P}(\widetilde{Y} = 1|Y = 0)$. The corresponding

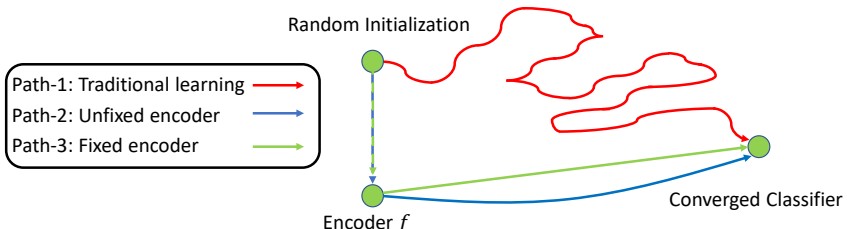

Figure 1: Illustration of different learning paths.

noisy dataset and distribution are denoted by $\widetilde{D} := \{(x_n, \tilde{y}_n)\}_{n \in [N]}$ and $\widetilde{\mathcal{D}}$. Define the expected risk of a classifier $C$ as $R(C) = \mathbb{E}_{\mathcal{D}}\left[\mathbb{1}(C(X) \neq Y)\right]$. The goal is to learn a classifier $C$ from the noisy distribution $\widetilde{\mathcal{D}}$ which also minimizes $R(C)$, i.e., learn the *Bayes optimal classifier* such that $C^{\text{Bayes}}(x) = \arg\max_{i \in \{0,1\}} \mathbb{P}(Y = i | X = x)$. For better presentation, we define the following notations: $X_+ = X|Y = 1, X_- = X|Y = 0$, and $X^{\text{clean}} = X|Y = \widetilde{Y}, X^{\text{noisy}} = X|Y \neq \widetilde{Y}$.

**Evaluation of SSL (Self-Supervised Learning):** SSL is usually evaluated by two steps: First, use SSL to train an encoder $f$ with only unlabeled data $X$, then add a linear classifier $g$ following the pre-trained encoder $f$ and only fine-tune $g$ on $(X, Y)$ with a fixed $f$. The high-level intuition is that, if the encoder $f$ is well learned by SSL, only fine-tuning linear classifier $g$ is often sufficient to achieve good performance on test data. If the test performance is comparable to SL (Supervised Learning), we call the gap between SSL and SL is small (Chen et al., 2020). Denote by $\mathcal{G}$ the space of linear classifier $g$. Fine-tuning linear layer $g$ on $(X, Y) \sim \mathcal{D}$ can be represented as:

$$\min_{g \in \mathcal{G}} \; \mathbb{E}_{\mathcal{D}}[\mathsf{CE}(g(f(X)), Y)],$$

where $\mathsf{CE}$ denotes the Cross-Entropy loss. Note the dimension of $f(X)$ is determined by the network structure, e.g., 512 for ResNet34. In binary classifications, $g(f(X)) \in [0, 1]$ where $g(f(X)) < 0.5$ indicates predicting class-0 and $g(f(X)) > 0.5$ corresponds to class-1. $g(f(X))$ is supposed to predict the same label as $C^{\text{Bayes}}(x)$.

## 3 ROBUSTNESS OF CROSS-ENTROPY WITH SSL FEATURES

We will analyze the robustness of Cross-Entropy with SSL features by comparing three different learning paths as illustrated in Figure 1. Path-1 is the traditional learning path that learns both encoder $f$ and linear classifier $g$ at the same time. Path-2 is the strategy applied in (Ghosh & Lan, 2021) that firstly pre-trains encoder $f$ with SSL, then treats the pre-trained model as a network initialization and jointly fine-tunes $f$ and $g$. Path-3 is an alternate SSL-based path that first learns the encoder $f$ then only fine-tunes the linear classifier $g$ with fixed $f$.

### 3.1 THEORETICAL TOOLS

We prepare some theoretical tools for our analyses. Our first theorem focuses on demonstrating the effectiveness of only fine-tuning linear classifier $g$ as in Path-3. We present Theorem 1 below.

**Theorem 1** *Let* $g_1 = \arg\min_{g \in \mathcal{G}} \mathbb{E}_{\mathcal{D}}[\mathsf{CE}(g(f(X)), Y)]$, $g_2 = \arg\min_{g \in \mathcal{G}} \mathbb{E}_{\widetilde{\mathcal{D}}}[\mathsf{CE}(g(f(X)), \widetilde{Y})]$. *Then if* $e_+ = e_- < 0.5$, *we have:*

$$\mathsf{Round}(g_1(f(X))) = \mathsf{Round}(g_2(f(X))) \tag{1}$$

*where $f$ is fixed encoder and $g$ is the linear classifier, $g(f(\cdot))$ denotes the output whose value ranges from 0 to 1. $\mathsf{Round}(p)$ is a predictor function that outputs 1 if $p > 0.5$ and outputs 0 otherwise.*

Theorem 1 shows with balanced error rates, simply fine-tuning a linear classifier $g$ on the noisy data distribution $\widetilde{\mathcal{D}}$ can achieve the same decision boundary as the optimal linear classifier obtained from the corresponding clean distribution $\mathcal{D}$. *i.e.,* $g_1(f)$ and $g_2(f)$ have the same predictions for all the samples. Theorem 1 can be generalized to the case with an arbitrary classifier beyond linear.

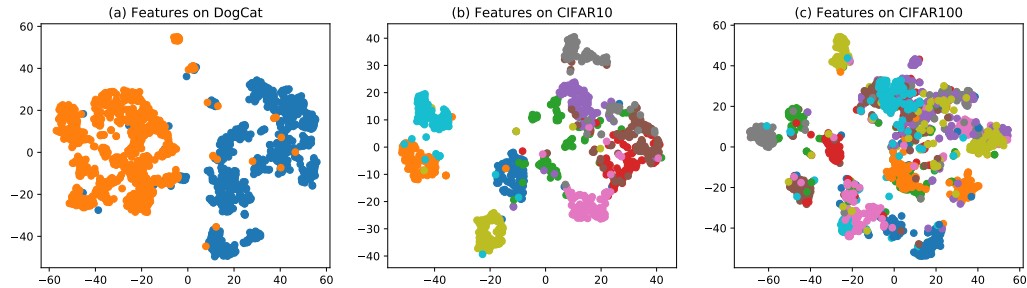

Figure 2: Tsne visualization of the self-supervised encoder outputs $f(X)$ on DogCat (Kaggle), CIFAR10 and CIFAR100 (Krizhevsky et al.). Different colors stand for different classes. SimCLR (Chen et al., 2020) is selected for SSL training. Quality of encoders: $(a) \succ (b) \succ (c)$.

However, admittedly with limited data, training complicated classifiers is hard to converge to the optimal decision boundary. We defer more details to the next subsection.

We then evaluate the former part of Path-3, i.e., the performance of SSL. Recall in Section 2, SSL is usually evaluated by performance gap between $f \circ g^{\text{Bayes}}$ and $C^{\text{Bayes}}$, where $g^{\text{Bayes}}$ is the optimal linear classifier trained on $\mathcal{D}$, $f \circ g^{\text{Bayes}}$ denotes the joint model given by $g^{\text{Bayes}}(f(X))$. We consider a tractable case in Assumption 1.

**Assumption 1** *The encoder outputs $f(X_+)$ and $f(X_-)$ follow Gaussian distribution with parameters $(\mu_1, \Sigma)$ and $(\mu_2, \Sigma)$, where $\Sigma = \sigma^2 \cdot I$, $I$ is the identity matrix.*

Assumption 1 states that the self-supervised features for each class follow simple Gaussian distributions. We check the effectiveness of this assumption by Figure 2. It can be observed that the features of each class may have overlaps, but a good SSL method is supposed to return features with good separations (small overlaps). In Assumption 1, we use $||\mu_1 - \mu_2||$ and $\sigma$ to capture the overlapping area of two classes. If $||\mu_1 - \mu_2||$ is large and $\sigma$ is small, then there exists small overlapping. Based on this assumption, we show the performance of SSL in Theorem 2.

**Theorem 2** *If $\mathbb{P}(Y = 1) = \mathbb{P}(Y = 0)$, the risk (error rate) of Bayes optimal classifier $f \circ g^{Bayes}$ follows as:*

$$R(f \circ g^{Bayes}) = 1 - \Phi\left(\frac{||\mu_1 - \mu_2||}{2 \cdot \sigma}\right) \tag{2}$$

*where $\Phi$ is the cumulative distribution function (CDF) of the standard Gaussian distribution.*

**Wrap-up** With Theorem 1, we know CE is robust, the performance of which is subject to $f$. Theorem 2 implies that if SSL features learned by $f$ exhibit good property, *i.e.,* when $||\mu_1 - \mu_2||$ is large and $\sigma$ is small, only fine-tuning $g$ can approach the Bayes optimal classifier $C^{\text{Bayes}}$. Therefore, good SSL features induce high performance. In summary, Theorem 1 and Theorem 2 connect SSL features with robustness and generalization ability of CE loss, providing an insight on why SSL features improve classification with label noise.

## 3.2 CAN WE FIX THE ENCODER?

We compare the performance of Path-2 with Path-3 in this subsection.

### 3.2.1 THEORETICAL ANALYSES

Denote by $\text{VC}(\mathcal{G})$ the VC-dimension of $\mathcal{G}$.

**Always fix good encoders** When we fine-tune $g$ on fixed $f$, the learning errors will come from two parts: the encoder error and the classification error. The encoder error is bounded in Theorem 2. Noting that $g \in \mathcal{G}$ is a linear classifier with relatively low VC-dimension and CE is a calibrated and convex function, the optimization can achieve global optimum with enough samples (Bartlett et al., 2006), e.g., w.p. $1 - \delta$, the learning error will be $\varepsilon$ with $\Theta(\frac{\text{VC}(\mathcal{G}) + \ln 1/\delta}{\varepsilon})$ *i.i.d.* instances (Vapnik,

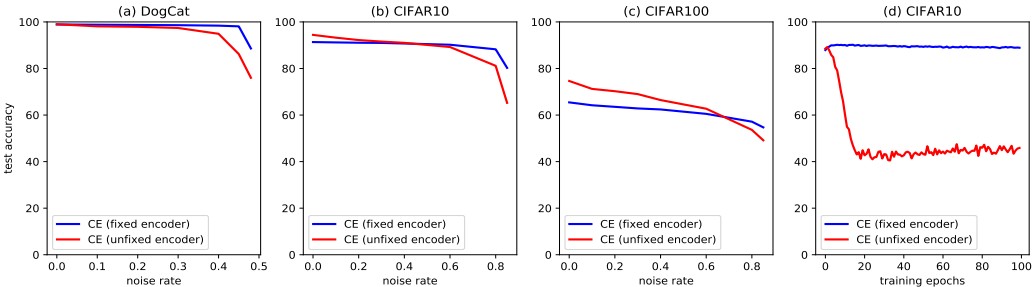

Figure 3: (a) (b) (c): Performance of CE on DogCat, CIFAR10 and CIFAR100 under symmetric noise rate. For each noise rate, the best epoch test accuracy is recorded. The blue line represents training with fixed encoder and the red line represents training with unfixed encoder; (d): test accuracy of CIFAR10 on each training epoch under symmetric 0.6 noise rate. We use ResNet50 for DogCat and ResNet34 for CIFAR10 and CIFAR100. SimCLR is used to deploy SSL pre-training. Detailed experimental setting are reported in the Appendix.

2013). Recall Theorem 1 also guarantees the optimality of $g$ learned under the noisy distribution $\widetilde{\mathcal{D}}$. Therefore, given a good encoder with large enough $\frac{\|\mu_1-\mu_2\|}{\sigma_1}$, we should keep it fixed since the simple linear classifier $g$ trained under label noise only requires $N = \Theta(\frac{\mathsf{VC}(\mathcal{G})+\ln 1/\delta}{\varepsilon})$ $i.i.d.$ instances in $\widetilde{D}$ to achieve around $\varepsilon$ errors with high probability, which performs well enough to approximate $C^{\mathsf{Bayes}}$.

**Updating $f$ is not stable** If $f$ is unfixed, due to the non-convexity and over-fitting capability of $f$, network trained on vanilla $\widetilde{D}$ will likely and ultimately overfit to noisy labels (Zhang et al., 2016), particularly when the number of instances is insufficient. For example, w.p. $1-\delta$, jointly learning $f$ and $g$ and achieving error $\varepsilon$ requires $\Theta(\frac{\mathsf{VC}(\mathcal{F}\circ\mathcal{G})+\ln 1/\delta}{\varepsilon})$ $i.i.d.$ instances, which is far more than the number required in only tuning $g$ due to the huge capacity of DNNs. Note the effective $i.i.d.$ instances would decrease with the increase of noise rates. Although a good initialization contributes to better convergence, $f$ will be misled by label noise in extreme cases. Therefore it is unreasonable to assume that CE will stably induce the network to approximate the Bayes optimal classifier under label noise. In other words, if $f$ is not fixed, due to the overfitting capability of $f$, the outputs of $f(X)$ would be contaminated by the label noise before $g$. In contrast, the original SSL features are independent from the label noise if $f$ is fixed. We therefore conclude that CE on vanilla $\widetilde{D}$ is not robust. Note Nodet et al. (2021) claimed the encoder should not be fixed during fine-tuning based on some experiments. We include more discussions in Appendix.

### 3.2.2 EMPIRICAL EVIDENCE

We conduct experiments in Figure 3 to support our theorems and analyses above.

**Overall performance** From Figure 3(a)–(c), when encoder $f$ is fixed, the performance of CE (fixed encoder) exhibits great robustness as noise rate increases and only has a little drop when the symmetric noise rate approximates the maximum theoretical value for each dataset (0.5 for DogCat, 0.9 for CIFAR10) which verifies Theorem 1 and Theorem 2. Note that the drop is because, to reach the condition specified in Equation (1), a substantial number of training samples are needed. However, for example, each class only has 500 samples in CIFAR100. As the noise rate increases, the number of clean samples will further decrease which makes the classifier hard to learn from the distribution.

**Fixed encoder benefits extreme label noise** Next, we observe the performance gap between SSL and SL when datasets are clean. This gap can be visualized in Figure 3 (a) (b) (c) at noise rate $= 0$. For DogCat , the gap is very small, thus CE with fixed encoder always exhibits better performance. For CIFAR10, the gap is moderate, the performances of CE (fixed encoder) and CE (unfixed encoder) are close when noise rate is small. However, when noise rate is high, CE with fixed encoder still exhibits better performance. For CIFAR100, the gap is large, thus CE with fixed encoder shows better performance only when noise rate is over 0.7.

**Fixed encoder has better convergence** Note that the performance in Figure 3 (a) (b) (c) are selected from best epoch accuracy. If we take the last epoch accuracy into consideration. CE with fixed encoder has great advantage over unfixed encoder. Figure 3 (d) depicts the performance of CE in

Table 1: Comparison of test accuracies of each method under instance-based label noise. CE (unfixed $f$ with random init.) is a common baseline adopted in the literature which trains a random initialized DNN using CE loss on noisy dataset.

| Method | Inst. CIFAR10 | | | Inst. CIFAR100 | | |
|---|---|---|---|---|---|---|
| | $\varepsilon = 0.2$ | $\varepsilon = 0.4$ | $\varepsilon = 0.6$ | $\varepsilon = 0.2$ | $\varepsilon = 0.4$ | $\varepsilon = 0.6$ |
| CORES (Cheng et al., 2021) | 89.50 | 82.84 | 79.66 | 61.25 | 47.81 | 37.85 |
| CE (unfixed $f$ with random init.) | 87.16 | 75.16 | 44.64 | 58.72 | 41.14 | 25.29 |
| CE (fixed $f$, no sampling) | 88.74 | 75.71 | 25.7 | 59.38 | 46.13 | 24.75 |
| CE (fixed $f$, down-sampling) | **90.12** | **84.19** | **82.06** | **62.88** | **61.1** | **58.93** |

terms of training epochs under 0.6 symmetric noise rate on CIFAR10. We find that even though these two approaches have almost same best epoch accuracy, CE with fixed encoder is more robust as training proceeds. For CE with unfixed encoder, since the self-supervised pre-trained network's weights serve as a better initialization, it shows good performance in the beginning. However, because of the memorizing effect of DNN, it will still over-fit to noisy labels in the end.

**Generalize to class- and instance-dependent label noise:** Theorem 1 shows that CE is robust to label noise when $e_- = e_+$. However, for a more general label noise such as asymmetric class-dependent label noise or instance-dependent label noise, CE with fixed encoder on vanilla $(f(X), \widetilde{Y})$ is hard to be guaranteed to be robust because for instance noise, noise rate varies for each class. But it does not mean SSL features can not help. We may still fix the encoder and perform a simple down-sampling to balance the noisy dataset before fine-tuning the linear classifier. The down-sampling is conducted to make $\mathbb{P}(\widetilde{Y} = i) = \mathbb{P}(\widetilde{Y} = j)$ in the noisy dataset which can well approximate the condition in Theorem 1 (Illustration is deferred to Appendix). Note that with down-sampling, the target distribution of clean labels also changes. However, Theorem 1 is proved without specifying the distribution of clean labels, suggesting that classifier is consistent even for imbalanced dataset under symmetric label noise. Secondly, with SSL features, classifier is robust to imbalanced imbalanced dataset (Yang & Xu, 2020). Thus down-sampling strategy to reduce error rate imbalances is helpful for other noise types beyond symmetric. We perform an experiment in Table 1 to verify the effectiveness of the down-sampling in the setting of instance-dependent label noise (generation of label noise is followed by (Cheng et al., 2021)). From the experiments, if no sampling strategy is conducted, CE with fixed encoder does not show many benefits. In some cases, it even performs worse than CE baseline (unfixed $f$ with random init). However, with down-sampling, CE with fixed encoder even outperforms CORES (Cheng et al., 2021) by a large margin. Even though our main purpose in the paper is not to develop new methods for achieving SOTA, we think this down-sampling strategy motivated by Theorem 1 can inspire further research on instance-dependent label noise by using SSL features.

## 3.3 TAKEAWAYS

Our both theoretical and empirical results demonstrate that: We need to simply fix the encoder when 1) the encoder is sufficiently good, 2) the high noise rate decreases the number of effective *i.i.d.* instances to a certain level, 3) the over-parameterized model makes it easy to overfit label noise. However, there are some concerns: 1) If the gap between SL and SSL is relatively large, CE with unfixed encoder has more advantage on small noise rate (e.g., Figure 3 (c)); 2) Consider an online classification system which always needs new training data to update itself. If we only fine-tune the linear classifier for keeping robustness to label noise, then due to the limited learnability of affine functions, this online classification system may learn very limited knowledge from unseen data. Therefore, it is necessary to update the encoder while keeping it stable and robust to label noise, which is the focus in the next section.

## 4 REGULARIZER FROM KD IMPROVES NETWORK ROBUSTNESS

In this section, we show that a regularizer from knowledge distillation (KD) (Park et al., 2019) can well alleviate the over-fitting problem by using the structure of SSL features. Different from classical KD settings where the dataset is clean, our analyses aim at regularizing classifiers learned with noisy

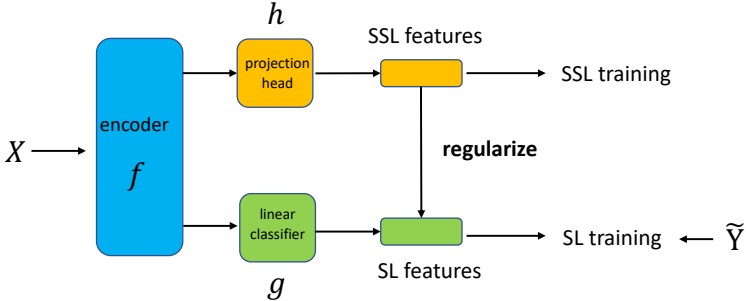

Figure 4: In this training framework, we adopt CE for SL training and InfoNCE for SSL training. During training, SSL features are utilized to perform regularization on SL features.

supervisions by SSL features. Understanding the working mechanism is important since it may contribute to the community a new perspective to learn with label noise by distilling information from SSL features. In the rest of this section, we will first provide theoretical understanding towards the regularizer under label noise, then perform experiments on different settings to test our analyses.

### 4.1 TRAINING FRAMEWORK

The training framework is shown in Figure 4, where a new learning path (SSL training) $f \rightarrow h$ is added to be parallel to Path-2 $f \rightarrow g$ (SL-training) in Figure 1. The newly added *projection head* $h$ is one-hidden-layer MLP (Multi Layer Perceptron) whose output represents SSL features (after dimension reduction). Its output is employed to regularize the output of linear classifier $g$ as $\text{Reg}(h(f(X)), g(f(X)))$. InfoNCE from SimCLR (Chen et al., 2020) is adopted for SSL training and CE is for SL training. InfoNCE and CE share a common encoder, inspired by the design of self distillation (Zhang et al., 2019). The loss function is defined as:

$$L = \mathbb{E}_{\widetilde{\mathcal{D}}} \left\{ \text{CE}(g(f(X)), \widetilde{Y}) + \text{InfoNCE}(h(f(X))) + \text{Reg}(h(f(X)), g(f(X))) \right\} \quad (3)$$

Intuitively, SL features is supposed to be benefited from the structure information from SSL features, e.g., clusterability (Zhu et al., 2021b) that instances with similar SSL features should have the same true label and instance with different SSL features should have different true labels. Mathematically, let $t_i = h(f(x_i))$, $s_i = g(f(x_i))$ and $\mathcal{X}^N$ to be the set of $N$ tuples of data samples. The distance between $t_i$ and $t_j$ can be represented as $\phi^w(t_i, t_j) = \frac{1}{m}\|t_i - t_j\|^w$, where $w \in \{1, 2\}$ and $m$ is a normalization term:

$$m = \frac{1}{|\mathcal{X}^2|} \sum_{(x_i, x_j) \in \mathcal{X}^2} \|t_i - t_j\|^w. \quad (4)$$

Then the expectation of $\text{Reg}(h(X), g(X))$ in Equation (3) can be estimated by:

$$\mathbb{E}_X \left[ \text{Reg}(h(X), g(X)) \right] \approx \frac{1}{|\mathcal{X}^2|} \sum_{(x_i, x_j) \in \mathcal{X}^2} d(\phi^w(t_i, t_j), \phi^w(s_i, s_j)) \quad (5)$$

where $d(\cdot)$ is a distance measure for two inputs. Popular choices are $l_1$, $l_2$ or square $l_2$ distance.

### 4.2 ANALYTICAL FRAMEWORK

Under the label noise setting, denote by $d_{i,j} = d(\phi^w(t_i, t_j), \phi^w(s_i, s_j))$, we have the following decomposition:

$$\frac{1}{|\mathcal{X}^2|} \sum_{(x_i, x_j) \in \mathcal{X}^2} d_{i,j} = \frac{1}{|\mathcal{X}^2|} \Big( \underbrace{\sum_{(x_i, x_j) \in \mathcal{X}^2_{\text{clean}}} d_{i,j}}_{\text{Term-1}} + \underbrace{\sum_{(x_i, x_j) \in \mathcal{X}^2_{\text{noisy}}} d_{i,j}}_{\text{Term-2}} + \underbrace{\sum_{x_i \in \mathcal{X}_{\text{clean}}, x_j \in \mathcal{X}_{\text{noisy}}} 2 \cdot d_{i,j}}_{\text{Term-3}} \Big). \quad (6)$$

where $\mathcal{X} = \mathcal{X}_{\text{clean}} \bigcup \mathcal{X}_{\text{noisy}}$, $\mathcal{X}_{\text{noisy}}$ denotes the set of samples whose labels are flipped. Note the regularizer mainly works when SSL features "disagree" with SL features, i.e., Term-3. For further analyses, we write Term-3 in the form of expectation with $d$ chosen as square $l_2$ distance, *i.e.*, MSE loss:

$$L_c = \mathbb{E}_{X^{\text{clean}}, X^{\text{noisy}}} \left( \frac{||g(f(X^{\text{clean}})) - g(f(X^{\text{noisy}}))||^1}{m_1} - \frac{||h(f(X^{\text{clean}})) - h(f(X^{\text{noisy}}))||^2}{m_2} \right)^2 \quad (7)$$

where $m_1$ and $m_2$ are normalization terms in Equation (4). Note in $L_c$, we use $w = 1$ for SL features and $w = 2$ for SSL features.[1] Denote the variance by $\text{var}(\cdot)$. Define notations:

$$X^{\text{noisy}}_+ := X|(\widetilde{Y} = 1, Y = 0), \quad X^{\text{noisy}}_- := X|(\widetilde{Y} = 0, Y = 1).$$

To theoretically measure and quantify how feature correction relates to network robustness, we make three assumptions as follows:

**Assumption 2 (Memorize clean instances)** $\forall n \in \{n|\tilde{y}_n = y_n\}, \text{CE}(g(f(x_n)), y_n) = 0.$

**Assumption 3 (Same overfitting)** $\text{var}(g(f(X^{noisy}_+))) = 0$ *and* $\text{var}(g(f(X^{noisy}_-))) = 0.$

**Assumption 4 (Gaussian-distributed SSL features)** *The SSL features follow Gaussian distributions, i.e., $h(f(X_{+1})) \sim \mathcal{N}(\mu_1, \Sigma)$ and $h(f(X_{-1})) \sim \mathcal{N}(\mu_2, \Sigma).$*

Assumption 2 implies that a DNN has confident predictions on clean samples. Assumption 3 implies that a DNN has the same degree of overfitting for each noisy sample. For example, an over-parameterized DNN can memorize all the noisy labels (Zhang et al., 2016; Liu, 2021). Thus these two assumptions are reasonable. Assumption 4 follows Assumption 1. Note in Figure 4, SSL features are from $h(f(X))$ rather than $f(X)$.

### 4.3 THEORETICAL UNDERSTANDING

Based on Assumptions 2–4, we present Theorem 3 to analyze the effect of $L_c$. Recall $X^{\text{noisy}}_+ := X|(\widetilde{Y} = 1, Y = 0), X^{\text{noisy}}_- := X|(\widetilde{Y} = 0, Y = 1).$

**Theorem 3** *When $e_- = e_+$ and $\mathbb{P}(Y = 1) = \mathbb{P}(Y = 0)$, minimizing $L_c$ respect to $g(f(X^{\text{noisy}}))$ on DNN results in the following solutions:*

$$\mathbb{E}_{X^{\text{noisy}}_+} \left[ g(f(X^{\text{noisy}}_+)) \right] = \frac{1}{2} - \frac{1}{2 + \Delta(\Sigma, \mu_1, \mu_2)}, \; \mathbb{E}_{X^{\text{noisy}}_-} \left[ g(f(X^{\text{noisy}}_-)) \right] = \frac{1}{2} + \frac{1}{2 + \Delta(\Sigma, \mu_1, \mu_2)},$$

*where $\Delta(\Sigma, \mu_1, \mu_2) := 8 \cdot tr(\Sigma)/||\mu_1 - \mu_2||^2$, $tr(\cdot)$ denotes the matrix trace, .*

Theorem 3 reveals a clean relationship between the quality of SSL features (given by $h(f(X))$) and the network robustness on noisy samples. Note the true label for $X^{\text{noisy}}_+$ is 0 and the true label for $X^{\text{noisy}}_-$ is 1. Theorem 3 shows the regularizer is leading the classifier to predict the feature with corrupted label to its corresponding true label, and the guidance is particular strong when $\Delta(\Sigma, \mu_1, \mu_2) \to 0$, e.g., $tr(\Sigma) \to 0$ or $||\mu_1 - \mu_2|| \to \infty$, such that $\mathbb{E}_{X^{\text{noisy}}_+}[g(f(X^{\text{noisy}}_+))] \to 0$ and $\mathbb{E}_{X^{\text{noisy}}_-}[g(f(X^{\text{noisy}}_-))] \to 1$. Beyond this ideal case, we discuss more interesting findings as:

- **Worst case:** The regularizer can always have a positive effect to avoid memorizing wrong labels as long as the model $f \circ h$ is better than randomly initialized models. This is because $\Delta(\Sigma, \mu_1, \mu_2) > 0$ when $\mu_1$ and $\mu_2$ are different, and the regularizer is leading the classifier to the true direction when $\Delta(\Sigma, \mu_1, \mu_2) > 0$.
- **Practical case:** The model $f \circ h$ is always better than randomly initialized models since SL training can guarantee the performance of encoder $f$ on clean instances, which may generalize to noisy instances, and SSL training can further improves the quality of $f \circ h$. Thus in practical cases, using SSL features to regularize learning with noisy labels is always beneficial.
- **More possibilities:** Note the proof of Theorem 3 does not rely on any SSL training process. From our first finding, we know any encoder helps if it is better than random models. This makes it possible to use some pre-trained encoders from other tasks.

**High-level understanding on structure regularization:** Even though we have built Theorem 3 to show SL features can benefit from the structure of SSL features by performing regularization, there

---

[1] Practically, different choices make negligible effects on performance. See more details in Appendix.

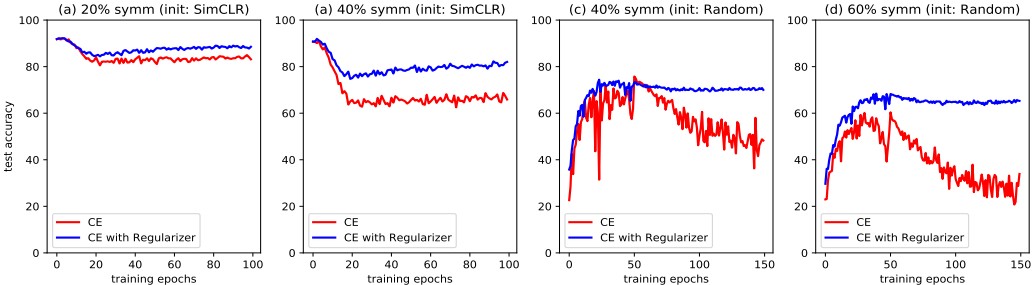

Figure 5: Experiments with respect to the regularizer on CIFAR10. ResNet34 is deployed for all the experiments. (a) (b): Encoder is pre-trained by SimCLR. Symmetric noise rate is 20% and 40%, respectively; (c) (d): Encoder is randomly initialized. Symmetric noise rate is 40% and 60%, respectively. The value of hyper-parameters and other detailed setting in the experiments are reported in the Appendix.

still lacks high-level understanding of what the regularization is exactly doing. Here we provide an insight in Theorem 4 which shows the regularization is implicitly maximizing mutual information between SL features and SSL features.

**Theorem 4** *Suppose there always exists a mapping $\xi$ to map $h(f(X))$ to $g(f(X))$. I.e., $g(f(X)) = \xi(h(f(X)))$. Then minimizing Equation (5) is implicitly maximizing Mutual Information between $h(f(X))$ and $g(f(X))$. I.e., when Equation (5) achieves $0$, the mutual information $I(h(f(X)), g(f(X)))$ achieves maximum.*

The above results facilitate a better understanding on what the regularizer is exactly doing. Note that Mutual Information itself has several popular estimators (Belghazi et al., 2018; Hjelm et al., 2018). It is a very interesting future direction to develop regularizes based on MI to perform regularization by utilizing SSL features.

## 4.4 EXPERIMENTS

We provide experiments on CIFAR10 to verify our theory and analysis. The overall experiments are shown in Figure 5. In the experiments, Regularizer is added at the very beginning since recent studies show that for random initialized network, the model tends to fit clean labels first (Arpit et al., 2017) and we hope the regularizer can improve the network robustness when DNN begins to fit noisy labels. From Figure 5 (c) (d), for CE training, the performance first increases then decreases since the network over-fits noisy labels as training proceeds. However, for CE with regularizer, the performance is more stable after it reaches the peak. For 60% noise rate, the peak point is also much higher than vanilla CE training. For Figure 5 (a) (b), since the network is not randomly initialized, it over-fits noisy labels at the very beginning and the performance gradually decreases. However, for CE with regularizer, it can help the network gradually increase the performance as the network reaches the lowest point (over-fitting state). This observation supports Theorem 3 that the regularizer can prevent DNN from over-fitting to noisy labels. More experiments and ablation studies can be found in the Appendix.

## 5 CONCLUSIONS AND DISCUSSIONS

In this paper, we have provided a theoretical understanding on why and how self-supervised features improve training from noisy labels and perform experiments to verify our analysis.

- We show that with SSL pre-trained encoder, CE itself is theoretically robust to symmetric label noise. With the down-sampling strategy, it can also exhibit great power on class- and instance-dependent label noise. This part answers **why** SSL features help (Theorem 1–2).
- We show that by utilizing SSL features, the regularizer in KD can help DNN alleviate the over-fitting problem on the noisy dataset and build theoretical insights on how SSL features relate to the network robustness. This part answers **how** SSL features help (Theorem 3–4).

Future directions can be done to develop more efficient sampling strategies for solving instance-based label noise with theoretical guarantees and develop efficient regularizers to combat label noise. The key for these future works is to leverage the nice structural property of SSL features.

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

APPENDIX

The Appendix is arranged as follows: Section A proves Theorem 1 and Theorem 2 in the main paper. Section B proves Theorem 3 and Theorem 4 in the main Paper. Section C illustrates why down-sampling can decrease the gap of noise rates. Section D provides additional discussions and experiments including further discussion of whether or not to fix the encoder when fine-tuning DNN on noisy dataset; analyses of why robust losses (Wang et al., 2019; Ma et al., 2020) may suffer severe performance degradation under high noise rates; the effect of distance measure in Equation (4) ($w$ = 1 or 2); ablation study in Section 4; visualization of features trained by vanilla CE loss; the effect of different SSL pre-trained methods; Experiments with human-annotated label noise. Section E elaborates the detailed experimental setting of all the experiments in the paper. Section F generalizes Theorem 1 to multi-class classification. Section G provides theoretical and experimental proof on why directly fine-tuning SSL features cannot be robust on asymmetric and instance label noise.

## A  PROOF FOR THEOREM 1,2

### A.1  PROOF FOR THEOREM 1

Define $g_1 = \min_g \mathbb{E}_{\mathcal{D}}[CE(g(f(X)), Y)]$ and $g_2 = \min_g \mathbb{E}_{\widetilde{\mathcal{D}}}[CE(g(f(X)), \widetilde{Y})]$. Suppose $g_1^{\text{Bayes}}$ and $g_2^{\text{Bayes}}$ are Bayes optimal classifiers under $\mathbb{P}(f(X), Y)$ and $\mathbb{P}(f(X), \widetilde{Y})$, respectively. Since CE is a calibrated and convex loss, with enough samples, $g_1$ and $g_2$ will converge to (approximate to) $g_1^{\text{Bayes}}$ and $g_2^{\text{Bayes}}$ (Bartlett et al., 2006). Thus to prove $g_1$ and $g_2$ have the same prediction for $f(X)$, it is identical to prove $g_1^{\text{Bayes}}$ and $g_2^{\text{Bayes}}$ have the same predictions for $f(X)$.

Let $p_+(\boldsymbol{x})$ be the distribution of $f(X_+)$ and $p_-(\boldsymbol{x})$ be the distribution of $f(X_-)$, according to the bayes rules:

$$\mathbb{P}(Y = 1|\boldsymbol{x}) = \frac{\mathbb{P}(Y = 1) \cdot p_+(\boldsymbol{x})}{\mathbb{P}(Y = 1) \cdot p_+(\boldsymbol{x}) + \mathbb{P}(Y = 0) \cdot p_-(\boldsymbol{x})} \tag{8}$$

$$\mathbb{P}(Y = 0|\boldsymbol{x}) = \frac{\mathbb{P}(Y = 0) \cdot p_-(\boldsymbol{x})}{\mathbb{P}(Y = 1) \cdot p_+(\boldsymbol{x}) + \mathbb{P}(Y = 0) \cdot p_-(\boldsymbol{x})} \tag{9}$$

Define $\psi(\boldsymbol{x}) = \mathbb{P}(Y = 1) \cdot p_+(\boldsymbol{x}) - \mathbb{P}(Y = 0) \cdot p_-(\boldsymbol{x})$. Thus $\psi(\boldsymbol{x}) = 0$ is the decision boundary for optimal Bayes classifier $g_1^{\text{Bayes}}$ on clean dataset. Since when $\psi(\boldsymbol{x}) > 0, \mathbb{P}(Y = 1|\boldsymbol{x}) > \mathbb{P}(Y = 0|\boldsymbol{x})$ from Equation (8) (9), the classifier will output the prediction as 1 and vice versa. Since the generation of label noise is independent to $f(X)$, the samples for each class whose labels are flipped also follows original distribution. Denote $p_+^e(\boldsymbol{x})$ to be the distribution of samples whose observed labels are 1 under label noise. Thus $p_+^e(\boldsymbol{x})$ and $p_-^e(\boldsymbol{x})$ can be represented as:

$$p_+^e(\boldsymbol{x}) = m_1 \cdot \{\mathbb{P}(Y = 1) \cdot (1 - e_+) \cdot p_+(\boldsymbol{x}) + \mathbb{P}(Y = 0) \cdot e_- \cdot p_-(\boldsymbol{x})\}$$
$$p_-^e(\boldsymbol{x}) = m_2 \cdot \{\mathbb{P}(Y = 1) \cdot e_+ \cdot p_+(\boldsymbol{x}) + \mathbb{P}(Y = 0) \cdot (1 - e_{-1}) \cdot p_-(\boldsymbol{x})\}$$

where $m_1$ and $m_2$ are normalization term to make distribution valid. Specifically, $m_1 = \frac{1}{\mathbb{P}(Y=1)\cdot(1-e_+)+\mathbb{P}(Y=0)\cdot e_-}$ and $m_2 = \frac{1}{\mathbb{P}(Y=1)\cdot e_+ +\mathbb{P}(Y=0)\cdot(1-e_-)}$. Under label noise, $\mathbb{P}(\widetilde{Y} = 1) = \mathbb{P}(Y = 1) \cdot (1 - e_+) + \mathbb{P}(Y = 0) \cdot e_-$ and $\mathbb{P}(\widetilde{Y} = 0) = \mathbb{P}(Y = 1) \cdot e_+ + \mathbb{P}(Y = 0) \cdot (1 - e_-)$. According bayes rules, we have:

$$\mathbb{P}(\widetilde{Y} = 1|\boldsymbol{x}) = \frac{\mathbb{P}(\widetilde{Y} = 1) \cdot p_+^e(\boldsymbol{x})}{\mathbb{P}(\widetilde{Y} = 1) \cdot p_+^e(\boldsymbol{x}) + \mathbb{P}(\widetilde{Y} = 0) \cdot p_-^e(\boldsymbol{x})} \tag{10}$$

$$\mathbb{P}(\widetilde{Y} = 0|\boldsymbol{x}) = \frac{\mathbb{P}(\widetilde{Y} = 0) \cdot p_-^e(\boldsymbol{x})}{\mathbb{P}(\widetilde{Y} = 1) \cdot p_+^e(\boldsymbol{x}) + \mathbb{P}(\widetilde{Y} = 0) \cdot p_-^e(\boldsymbol{x})} \tag{11}$$

Without loss of generality, we assume there exits a $\boldsymbol{x}_0$ in the feature space which satisfy $\psi(\boldsymbol{x}_0) > 0$. Thus for clean dataset, $\mathbb{P}(Y = 1) \cdot p_+(\boldsymbol{x}) > \mathbb{P}(Y = 0) \cdot p_-(\boldsymbol{x})$ and $\mathbb{P}(Y = 1|\boldsymbol{x}_0) > \mathbb{P}(Y = 0|\boldsymbol{x}_0)$. We aim to show that for noisy dataset, $\mathbb{P}(\widetilde{Y} = 1|\boldsymbol{x}_0) > \mathbb{P}(\widetilde{Y} = 0|\boldsymbol{x}_0)$, which means $\psi(\boldsymbol{x}) = 0$ is also the decision boundary that gives Bayes optimal classifier $g_2^{Bayes}$ under label noise.

According to Equation (10) and Equation (11),

$$\begin{aligned}
&\mathbb{P}(\widetilde{Y} = 1|\boldsymbol{x}_0) - \mathbb{P}(\widetilde{Y} = 0|\boldsymbol{x}_0) \\
&= \frac{(1 - 2 \cdot e_+) \cdot \mathbb{P}(Y = 1) \cdot p_+(\boldsymbol{x}_0) - (1 - 2 \cdot e_-) \cdot \mathbb{P}(Y = 0) \cdot p_-(\boldsymbol{x}_0)}{\text{normalization}}
\end{aligned} \tag{12}$$

where normalization $= \mathbb{P}(\widetilde{Y} = 1) \cdot p_+^e(\boldsymbol{x}) + \mathbb{P}(\widetilde{Y} = 0) \cdot p_-^e(\boldsymbol{x}) > 0$. Since $\mathbb{P}(Y = 1) \cdot p_+(\boldsymbol{x}_0) > \mathbb{P}(Y = 0) \cdot p_-(\boldsymbol{x}_0)$, if $e_+ = e_- < 0.5$, we have $\mathbb{P}(\widetilde{Y} = 1|\boldsymbol{x}_0) - \mathbb{P}(\widetilde{Y} = 0|\boldsymbol{x}_0) > 0$. Thus $\psi(\boldsymbol{x}) = 0$ is also the decision boundary of $g_2^{\text{Bayes}}$. $g_1^{\text{Bayes}}$ and $g_2^{\text{Bayes}}$ will output the same prediction for all the samples.

Proof done.

## A.2 PROOF FOR THEOREM 2

Let $d = ||\mu_1 - \mu_2||$. Without loss of generality, we assume $\mu_1 = (-\frac{d}{2}, 0, \ldots, 0)$ and $\mu_2 = (\frac{d}{2}, 0, \ldots, 0)$ (This can be done by applying affine transformation on original coordinates without changing actual positions of $f(X_+)$ and $f(X_-)$). Under the distribution of Gaussian assumption, we write the distribution of $f(X_+)$ as follows:

$$p_+(\boldsymbol{x}) = \frac{1}{(2\pi)^{\frac{k}{2}} \sigma^k} \exp\{-\frac{1}{2\sigma^2}(\boldsymbol{x} - \mu_1)^T(\boldsymbol{x} - \mu_1)\} \tag{13}$$

where $\boldsymbol{x} = (x_1, \ldots, x_k)^T \in \mathbb{R}^k$, $x_k$ denotes the variable of $\boldsymbol{x}$ in the $k$-th dimension. Similarly, the pdf of $f(X_-)$ can be represented as:

$$p_-(\boldsymbol{x}) = \frac{1}{(2\pi)^{\frac{k}{2}} \sigma^k} \exp\{-\frac{1}{2\sigma^2}(\boldsymbol{x} - \mu_2)^T(\boldsymbol{x} - \mu_2)\} \tag{14}$$

Following notations from A.1, solving $\psi(\boldsymbol{x}) = 0$ under the conditions of $\mathbb{P}(Y = 1) = \mathbb{P}(Y = 0)$ gives $x_1 = 0$ ($x_1$ is the variable of $\boldsymbol{x}$ in the first dimension). Thus, when $x_1 < 0$, the Bayes optimal classifier $f \circ g^{\text{Bayes}}$ outputs prediction as 1. *I.e.,* for samples whose labels are 1, $f \circ g^{\text{Bayes}}$ gives right prediction when $x_1 < 0$ and wrong prediction when $x_1 > 0$. The risk of of $f \circ g^{\text{Bayes}}$ can be written as:

$$\begin{aligned}
R(f \circ g^{\text{Bayes}}) &= \mathbb{E}_{X,Y} \mathbb{1}(f \circ g^{\text{Bayes}}(X) \neq Y) \\
&= \mathbb{E}_Y \mathbb{E}_{X|Y} \mathbb{1}(f \circ g^{\text{Bayes}}(X) \neq Y) \\
&= \mathbb{P}(Y = 1) \cdot \mathbb{E}_{X|Y=1} \mathbb{1}(f \circ g^{\text{Bayes}}(X) \neq 1) + \mathbb{P}(Y = 0) \cdot \mathbb{E}_{X|Y=0} \mathbb{1}(f \circ g^{\text{Bayes}}(X) \neq 0) \\
&= \frac{1}{2} \cdot \mathbb{E}_{X|Y=1} \mathbb{1}(f \circ g^{\text{Bayes}}(X) \neq 1) + \frac{1}{2} \cdot \mathbb{E}_{X|Y=0} \mathbb{1}(f \circ g^{\text{Bayes}}(X) \neq 0) \\
&\overset{(a)}{=} \mathbb{E}_{X|Y=1} \mathbb{1}(f \circ g^{\text{Bayes}}(X) \neq 1) \\
&= \left\{ \int_{\boldsymbol{x}|x_1<0} p_+(\boldsymbol{x})\mathrm{d}\boldsymbol{x} \right\} \cdot 0 + \left\{ \int_{\boldsymbol{x}|x_1>0} p_+(\boldsymbol{x})\mathrm{d}\boldsymbol{x} \right\} \cdot 1 \\
&\overset{(b)}{=} 1 - \Phi(\frac{d}{2 \cdot \sigma})
\end{aligned}$$

(a) is satisfied because of the symmetry of $p_+(\boldsymbol{x})$ and $p_-(\boldsymbol{x})$. We derive (b) as follows:

$$\int_{\boldsymbol{x}|x_1<0} p_+(\boldsymbol{x})\mathrm{d}\boldsymbol{x}$$

$$= \int_{\boldsymbol{x}|x_1<0} \frac{1}{(2\pi)^{\frac{k}{2}}\sigma^k} \exp\{-\frac{1}{2\sigma^2}(\boldsymbol{x}-\mu_1)^T(\boldsymbol{x}-\mu_1)\}\mathrm{d}\boldsymbol{x}$$

$$= \int_{\boldsymbol{x}|x_1<0} \frac{1}{(2\pi)^{\frac{k}{2}}\sigma^k} \exp\{-\frac{(x_1+\frac{d}{2})^2 + x_2^2 + \cdots + x_k^2}{2\sigma^2}\mathrm{d}\boldsymbol{x}$$

$$= \int_{x_2} \frac{1}{\sqrt{2\pi}\sigma}\exp\{-\frac{x_2^2}{2\sigma^2}\}\mathrm{d}x_2 \cdots \int_{x_k} \frac{1}{\sqrt{2\pi}\sigma}\exp\{-\frac{x_k^2}{2\sigma^2}\}\mathrm{d}x_k \int_{x_1<0} \frac{1}{\sqrt{2\pi}\sigma}\exp\{-\frac{(x_1+\frac{d}{2})^2}{2\sigma^2}\}\mathrm{d}x_1$$

$$= 1 \cdot 1 \cdots \cdot \int_{x_1<0} \frac{1}{\sqrt{2\pi}\sigma}\exp\{-\frac{(x_1+\frac{d}{2})^2}{2\sigma^2}\}\mathrm{d}x_1$$

Since $\int_{x_1} \frac{1}{\sqrt{2\pi}\sigma}\exp\{-\frac{(x_1+\frac{d}{2})^2}{2\sigma^2}\}\mathrm{d}x_1$ is one dimensional gaussian with mean $-\frac{d}{2}$ and deviation $\sigma$, it is easy to calculate that $\int_{x_1<0} \frac{1}{\sqrt{2\pi}\sigma}\exp\{-\frac{(x_1+\frac{d}{2})^2}{2\sigma^2}\}\mathrm{d}x_1 = \Phi(\frac{d}{2\cdot\sigma})$, where $\Phi$ is the cdf of standard gaussian distribution.

Thus $\int_{x_1>0} \frac{1}{\sqrt{2\pi}\sigma}\exp\{-\frac{(x_1+\frac{d}{2})^2}{2\sigma^2}\}\mathrm{d}x_1 = 1 - \int_{x_1<0} \frac{1}{\sqrt{2\pi}\sigma}\exp\{-\frac{(x_1+\frac{d}{2})^2}{2\sigma^2}\}\mathrm{d}x_1 = 1 - \Phi(\frac{d}{2\cdot\sigma})$ and Equation (b) is satisfied.

Proof done.

## B  PROOF FOR THEOREM 3,4

**Lemma 1** *If $X$ and $Y$ are independent and follow gaussian distribution: $X \sim \mathcal{N}(\mu_X, \Sigma_X)$ and $Y \sim \mathcal{N}(\mu_Y, \Sigma_Y)$, Then: $\mathbb{E}_{X,Y}(||X-Y||^2) = ||\mu_X - \mu_Y||^2 + tr(\Sigma_X + \Sigma_Y)$.*

### B.1  PROOF FOR THEOREM 3

Before the derivation, we define some notations for better presentation. Define the labels of $X^{\text{clean}}$ as $Y^{\text{clean}}$ and the labels of $X^{\text{noisy}}$ as $Y^{\text{noisy}}$. Under the label noise, it is easy to verify $\mathbb{P}(Y^{\text{clean}} = 1) = \frac{\mathbb{P}(Y=1)\cdot(1-e_+)}{\mathbb{P}(Y=1)\cdot(1-e_+)+\mathbb{P}(Y=0)\cdot(1-e_-)}$ and $\mathbb{P}(Y^{\text{noisy}} = 1) = \frac{\mathbb{P}(Y=0)\cdot e_-}{\mathbb{P}(Y=0)\cdot e_-+\mathbb{P}(Y=1)\cdot e_+}$. Let $p_1 = \mathbb{P}(Y^{\text{clean}} = 1)$, $p_2 = \mathbb{P}(Y^{\text{noisy}} = 1)$, $g(f(X))$ and $h(f(X))$ to be simplified as $gf(X)$ and $hf(X)$.

In the case of binary classification, $gf(x)$ is one dimensional value which denotes the network prediction on $x$ belonging to $Y = 1$. $L_c$ can be written as:

$$\mathbb{E}_{X^{\text{clean}},X^{\text{noisy}}} \underbrace{\left(\frac{||gf(X^{\text{clean}}) - gf(X^{\text{noisy}}))||^1}{m_1} - \frac{||hf(X^{\text{clean}}) - hf(X^{\text{noisy}})||^2}{m_2}\right)^2}_{\text{denoted as} \quad \Psi(X^{\text{clean}},X^{\text{noisy}})}$$

$$\overset{(a)}{=} \mathbb{E}_{\substack{(X^{\text{clean}},Y^{\text{clean}}) \\ (X^{\text{noisy}},Y^{\text{noisy}})}} \Psi(X^{\text{clean}}, X^{\text{noisy}})$$

$$= p_1 \cdot p_2 \cdot \mathbb{E}_{X_+^{\text{clean}},X_+^{\text{noisy}}} \Psi(X_+^{\text{clean}}, X_+^{\text{noisy}}) + (1-p_1) \cdot p_2 \cdot \mathbb{E}_{X_-^{\text{clean}},X_+^{\text{noisy}}} \Psi(X_-^{\text{clean}}, X_+^{\text{noisy}})$$

$$+ p_1 \cdot (1-p_2) \cdot \mathbb{E}_{X_+^{\text{clean}},X_-^{\text{noisy}}} \Psi(X_+^{\text{clean}}, X_-^{\text{noisy}}) + (1-p_1) \cdot (1-p_2) \cdot \mathbb{E}_{X_-^{\text{clean}},X_-^{\text{noisy}}} \Psi(X_-^{\text{clean}}, X_-^{\text{noisy}})$$

where $m_1$ and $m_2$ are normalization terms from Equation (4).  (a) is satisfied because $\Psi(X^{\text{clean}}, X^{\text{noisy}})$ is irrelevant to the labels. We derive $\Psi(X_+^{\text{clean}}, X_+^{\text{noisy}})$ as follows:

$$\mathbb{E}_{X_+^{\text{clean}}, X_+^{\text{noisy}}} \Psi(X_+^{\text{clean}}, X_+^{\text{noisy}})$$

$$\overset{(b)}{=} \mathbb{E}_{X_+^{\text{clean}}, X_+^{\text{noisy}}} \left( \frac{||1 - gf(X_+^{\text{noisy}})||^1}{m_1} - \frac{||hf(X_+^{\text{clean}}) - hf(X_+^{\text{noisy}})||^2}{m_2} \right)^2$$

$$\overset{(c)}{=} \mathbb{E}_{X_+^{\text{clean}}, X_+^{\text{noisy}}} \left( \frac{1 - gf(X_+^{\text{noisy}})}{m_1} - \frac{||hf(X_+^{\text{clean}}) - hf(X_+^{\text{noisy}})||^2}{m_2} \right)^2$$

$$\overset{(d)}{=} \mathbb{E}_{X_+^{\text{clean}}, X_+^{\text{noisy}}} \left( \frac{gf(X_+^{\text{noisy}})}{m_1} - \left( \frac{1}{m_1} - \frac{||hf(X_+^{\text{clean}}) - hf(X_+^{\text{noisy}})||^2}{m_2} \right) \right)^2$$

(b) is satisfied because from Assumption 2, DNN has confident prediction on clean samples. (c) is satisfied because $g(f(x))$ is one dimensional value which ranges from 0 to 1. From Assumption 4, $hf(X_+)$ and $hf(X_-)$ follows gaussian distribution with parameter $(\mu_1, \Sigma)$ and $(\mu_2, \Sigma)$. Thus according to Lemma 1, we have $\mathbb{E}_{X_+^{\text{clean}}, X_+^{\text{noisy}}} ||hf(X_+^{\text{clean}}) - hf(X_+^{\text{noisy}})||^2 = ||\mu_1 - \mu_2||^2 + 2 \cdot tr(\Sigma)$. Similarly, one can calculate $\mathbb{E}_{X_-^{\text{clean}}, X_+^{\text{noisy}}} ||hf(X_-^{\text{clean}}) - hf(X_+^{\text{noisy}})||^2 = 2 \cdot tr(\Sigma)$. It can be seen that (d) is function with respect to $gf(X_+^{\text{noisy}})$. Similarly, $\Psi(X_-^{\text{clean}}, X_+^{\text{noisy}})$ is also a function with respect to $gf(X_+^{\text{noisy}})$ while $\Psi(X_+^{\text{clean}}, X_-^{\text{noisy}})$ and $\Psi(X_-^{\text{clean}}, X_-^{\text{noisy}})$ are functions with respect to $gf(X_-^{\text{noisy}})$. Denote $d(+, +) = \mathbb{E}_{X_+^{\text{clean}}, X_+^{\text{noisy}}} ||hf(X_+^{\text{clean}}) - hf(X_+^{\text{noisy}})||^2$. After organizing $\Psi(X_+^{\text{clean}}, X_+^{\text{noisy}})$ and $\Psi(X_-^{\text{clean}}, X_+^{\text{noisy}})$, we have:

$$\min_{gf(X_+^{\text{noisy}})} p_1 \cdot p_2 \cdot \mathbb{E}_{X_+^{\text{clean}}, X_+^{\text{noisy}}} \Psi(X_+^{\text{clean}}, X_+^{\text{noisy}}) + (1 - p_1) \cdot p_2 \cdot \mathbb{E}_{X_-^{\text{clean}}, X_+^{\text{noisy}}} \Psi(X_-^{\text{clean}}, X_+^{\text{noisy}})$$

$$\Rightarrow \min_{gf(X_+^{\text{noisy}})} (\mathbb{E}_{X_+^{\text{noisy}}} gf(X_+^{\text{noisy}}))^2$$

$$- \left( 2 \cdot p_1 \left( 1 - \frac{m_1 \cdot d(+, +)}{m_2} \right) + 2 \cdot (1 - p_1) \left( \frac{m_1 \cdot d(-, +)}{m_2} \right) \right) \cdot \mathbb{E}_{X_+^{\text{noisy}}} gf(X_+^{\text{noisy}})$$

$$+ \text{ constant with respect to } gf(X_+^{\text{noisy}})$$

(15)

Note in Equation (15), we use $(\mathbb{E}_{X_+^{\text{noisy}}} gf(X_+^{\text{noisy}}))^2$ to approximate $\mathbb{E}_{X_+^{\text{noisy}}} gf(X_+^{\text{noisy}})^2$ since from Assumption 3, $\text{var}(g(f(X_+^{\text{noisy}}))) \to 0$. Now we calculate $m_1$ and $m_2$ from Equation (4):

$$m_1 = p_1 \cdot p_2 \cdot (1 - \mathbb{E}_{X_+^{\text{noisy}}} gf(X_+^{\text{noisy}})) + (1 - p_1) \cdot p_2 \cdot \mathbb{E}_{X_+^{\text{noisy}}} gf(X_+^{\text{noisy}})$$

$$+ p_1 \cdot (1 - p_2) \cdot (1 - \mathbb{E}_{X_-^{\text{noisy}}} gf(X_-^{\text{noisy}})) + (1 - p_1) \cdot (1 - p_2) \cdot \mathbb{E}_{X_-^{\text{noisy}}} gf(X_-^{\text{noisy}})$$

(16)

$$m_2 = p_1 \cdot p_2 \cdot d(+, +) + (1 - p_1) \cdot p_2 \cdot d(-, +) + p_1 \cdot (1 - p_2) \cdot d(+, -) + (1 - p_1)(1 - p_2) \cdot d(-, -)$$

Under the condition of $\mathbb{P}(Y = 1) = \mathbb{P}(Y = 0)$, $e_- = e_+$, we have $p_1 = p_2 = \frac{1}{2}$, $m_2 = \frac{4 \cdot tr(\Sigma) + ||\mu_1 - \mu_2||^2}{2}$, $m_1 = \frac{1}{2}$, which is constant with respect to $\mathbb{E}_{X_+^{\text{noisy}}} gf(X_+^{\text{noisy}})$ and $\mathbb{E}_{X_-^{\text{noisy}}} gf(X_-^{\text{noisy}})$ in Equation (16). Thus Equation (15) is a quadratic equation with respect to $\mathbb{E}_{X_+^{\text{noisy}}} gf(X_+^{\text{noisy}})$. Then when Equation (15) achieves global minimum, we have:

$$\mathbb{E}_{X_+^{\text{noisy}}} gf(X_+^{\text{noisy}}) = p_1 - \frac{m_1}{m_2} (p_1 \cdot d(+, +) - (1 - p_1) \cdot d(-, +))$$

$$= \frac{1}{2} - \frac{1}{2 + \frac{8 \cdot tr(\Sigma)}{||\mu_1 - \mu_2||^2}}$$

(17)

Similarly, organizing $\Psi(X_+^{\text{clean}}, X_-^{\text{noisy}})$ and $\Psi(X_-^{\text{clean}}, X_-^{\text{noisy}})$ gives the solution of $\mathbb{E}_{X_-^{\text{noisy}}} gf(X_-^{\text{noisy}})$:

$$\mathbb{E}_{X^{\text{noisy}}_-} g f(X^{\text{noisy}}_-) = p_1 + \frac{m_1}{m_2}(p_1 \cdot d(-,-) - (1-p_1) \cdot d(+,-))$$
$$= \frac{1}{2} + \frac{1}{2 + \frac{8 \cdot tr(\Sigma)}{||\mu_1 - \mu_2||^2}} \tag{18}$$

Proof Done.

## B.2 PROOF FOR THEOREM 4

We first refer to a property of Mutual Information:

$$I(X; Y) = I(\psi(X); \phi(Y)) \tag{19}$$

where $\psi$ and $\phi$ are any invertible functions. This property shows that mutual information is invariant to invertible transformations (Cover, 1999). Thus to prove the theorem, we only need to prove that $\xi$ in Theorem 4 must be an invertible function when Equation (5) is minimized to 0. Since when $\xi$ is invertible, $I(h(f(X)), g(f(X))) = I(h(f(X)), \xi(h(f(X)))) = I(h(f(X)), h(f(X)))$.

We prove this by contradiction.

Let $t_i = h(f(x_i))$ and $s_i = g(f(x_i))$. Suppose $\xi$ is not invertible, then there must exists $s_i$ and $s_j$ where $s_i \neq s_j$ which satisfy $t_j = \xi(s_i) = t_i$. However, under this condition, $t_i - t_j = 0$ and $s_i - s_j \neq 0$, Equation (5) can not be minimized to 0. Thus when Equation (5) is minimized to 0, $\xi$ must be an invertible function.

Proof done.

## B.3 PROOF FOR LEMMA 1

By the independence condition, $Z = X - Y$ also follows gaussian distribution with parameter $(\mu_X - \mu_Y, \Sigma_X + \Sigma_Y)$.

Write $Z$ as $Z = \mu + LU$ where $U$ is a standard gaussian and $\mu = \mu_X - \mu_Y$, $LL^T = \Sigma_X + \Sigma_Y$. Thus

$$||Z||^2 = Z^T Z = \mu^T \mu + \mu^T LU + U^T L^T \mu + U^T L^T LU \tag{20}$$

Since $U$ is standard gaussian, $\mathbb{E}(U) = \mathbf{0}$. We have

$$\mathbb{E}(||Z||^2) = \mu^T \mu + \mathbb{E}(U^T L^T LU)$$
$$= \mu^T \mu + \mathbb{E}(\sum_{k,l}(L^T L)_{k,l} U_k U_l)$$
$$\overset{(a)}{=} \mu^T \mu + \sum_k (L^T L)_{k,k} \tag{21}$$
$$= \mu^T \mu + tr(L^T L)$$
$$= ||\mu_X - \mu_Y||^2 + tr(\Sigma_X + \Sigma_Y)$$

(a) is satisfied because $U$ is standard gaussian, thus $\mathbb{E}(U_k^2) = 1$ and $\mathbb{E}(U_k U_l) = 0$ $(k \neq l)$.

Proof Done.

## C ILLUSTRATING DOWN-SAMPLING STRATEGY

We illustrate in the case of binary classification with $e_+ + e_- < 1$. Suppose the dataset is balanced, at the initial sate, $e_+ > e_-$. After down-sampling, the noise rate becomes $e_+^*$ and $e_-^*$. We aim to prove two propositions:

**Proposition 1** *If $e_+$ and $e_-$ are known, the optimal down-sampling rate can be calculated by $e_+$ and $e_-$ to make $e_+^* = e_-^*$*

**Proposition 2** *If $e_+$ and $e_-$ are not known. When down-sampling strategy is to make $\mathbb{P}(\widetilde{Y} = 1) = \mathbb{P}(\widetilde{Y} = 0)$, then $0 < e_+^* - e_-^* < e_+ - e_-$.*

*Proof for Proposition 1:* Since dataset is balanced with initial $e_+ > e_-$, we have $\mathbb{P}(\widetilde{Y} = 1) < \mathbb{P}(\widetilde{Y} = 0)$. Thus down-sampling is conducted at samples whose observed label are 0. Suppose the random down-sampling rate is $r$, then $e_+^* = \frac{r \cdot e_+}{1 - e_+ + r \cdot e_+}$ and $e_-^* = \frac{e_-}{r \cdot (1 - e_-) + e_-}$. If $e_+^* = e_-^*$, we have:

$$\frac{r \cdot e_+}{1 - e_+ + r \cdot e_+} = \frac{e_-}{r \cdot (1 - e_-) + e_-} \tag{22}$$

Thus the optimal down-sampling rate $r = \sqrt{\frac{e_- \cdot (1 - e_+)}{e_+ \cdot (1 - e_-)}}$, which can be calculated if $e_-$ and $e_+$ are known.

*Proof for Proposition 2:* If down sampling strategy is to make $\mathbb{P}(\widetilde{Y} = 1) = \mathbb{P}(\widetilde{Y} = 0)$, then $r \cdot (e_+ + 1 - e_-) = 1 - e_+ + e_-$, we have $r = \frac{1 - e_+ + e_-}{1 - e_- + e_+}$. Thus $e_+^*$ can be calculated as:

$$e_+^* = \frac{r \cdot e_+}{1 - e_+ + r \cdot e_+}$$
$$= \frac{(1 - e_+ + e_-) \cdot e_+}{(1 - e_+) \cdot (1 - e_- + e_+) + e_+ \cdot (1 - e_+ + e_-)}$$

Denote $\alpha = \frac{1 - e_+ + e_-}{(1 - e_+) \cdot (1 - e_- + e_+) + e_+ \cdot (1 - e_+ + e_-)}$. Since $e_+ > e_-$, $1 - e_- + e_+ > 1 - e_+ + e_-$, $\alpha = \frac{1 - e_+ + e_-}{(1 - e_+) \cdot (1 - e_- + e_+) + e_+ \cdot (1 - e_+ + e_-)} < \frac{1 - e_+ + e_-}{(1 - e_+) \cdot (1 - e_+ + e_-) + e_+ \cdot (1 - e_+ + e_-)} = 1$.

Similarly, $e_-^*$ can be calculated as:

$$e_-^* = \frac{e_-}{e_- + r \cdot (1 - e_-)}$$
$$= \frac{(1 - e_- + e_+) \cdot e_-}{e_- \cdot (1 - e_- + e_+) + (1 - e_-) \cdot (1 - e_+ + e_-)}$$

Denote $\beta = \frac{1 - e_- + e_+}{e_- \cdot (1 - e_- + e_+) + (1 - e_-) \cdot (1 - e_+ + e_-)}$. Since $e_+ > e_-$, $1 - e_- + e_+ > 1 - e_+ + e_-$, $\beta = \frac{1 - e_- + e_+}{e_- \cdot (1 - e_- + e_+) + (1 - e_-) \cdot (1 - e_+ + e_-)} > \frac{1 - e_- + e_+}{e_- \cdot (1 - e_- + e_+) + (1 - e_-) \cdot (1 - e_- + e_+)} = 1$. Since $\alpha \cdot e_+ < e_+$ and $\beta \cdot e_- > e_-$, we have $e_+^* - e_-^* = \alpha \cdot e_+ - \beta \cdot e_- < e_+ - e_-$.

Next, we prove $e_+^* > e_-^*$, following the derivation below:

$$e_+^* > e_-^*$$
$$\implies \frac{r \cdot e_+}{1 - e_+ + r \cdot e_+} > \frac{e_-}{e_- + r \cdot (1 - e_-)}$$
$$\implies r > \sqrt{\frac{e_- \cdot (1 - e_+)}{e_+ \cdot (1 - e_-)}}$$
$$\implies \frac{1 - e_+ + e_-}{1 - e_- + e_+} > \sqrt{\frac{e_- \cdot (1 - e_+)}{e_+ \cdot (1 - e_-)}} \tag{23}$$
$$\implies e_+ \cdot (1 - e_+) + \frac{e_+ \cdot e_-^2}{1 - e_+} > e_- \cdot (1 - e_-) + \frac{e_- \cdot e_+^2}{1 - e_-}$$

Let $f(e_+) = e_+ \cdot (1 - e_+) + \frac{e_+ \cdot e_-^2}{1 - e_+} - e_- \cdot (1 - e_-) - \frac{e_- \cdot e_+^2}{1 - e_-}$. Since we have assumed $e_- < e_+$ and $e_- + e_+ < 1$. Thus proving $e_+^* > e_-^*$ is identical to prove $f(e_+) > 0$ when $e_- < e_+ < 1 - e_-$.

Firstly, it is easy to verify when $e_+ = e_-$ or $e_+ = 1 - e_-$, $f(e_+) = 0$. From Mean Value Theory, there must exists a point $e_0$ which satisfy $f'(e_0) = 0$ where $e_+ < e_0 < 1 - e_-$. Next, we differentiate

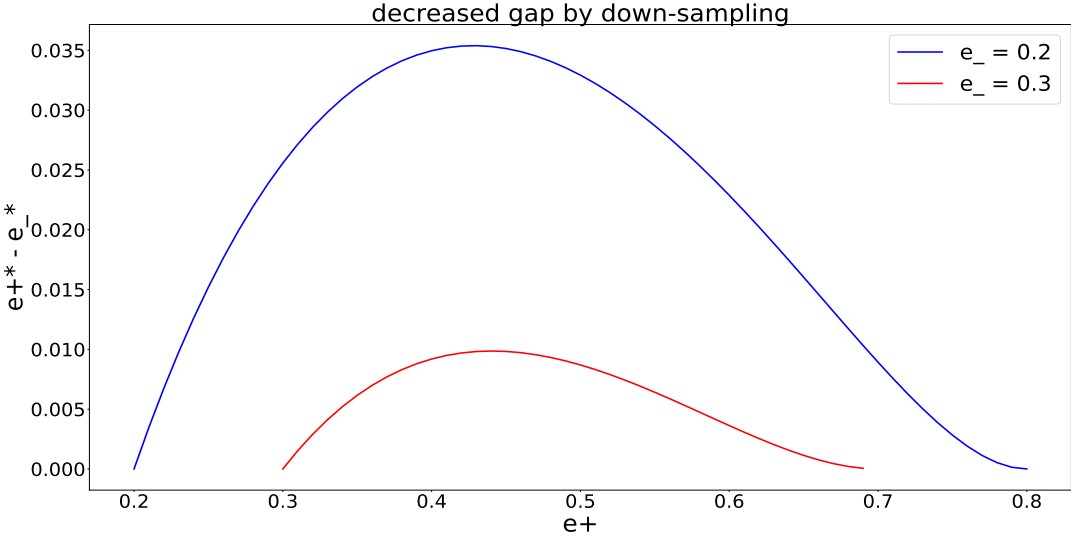

Figure 6: Visualizing decreased gap by down-sampling strategy.

$f(e_+)$ as follows:

$$f'(e_+) = \frac{(1-e_+)^2 \cdot (1-e_-) + e_-^2 \cdot (1-e_-) - 2 \cdot e_+(1-e_+)^2}{(1-e_+)^2 \cdot (1-e_-)} \tag{24}$$

It can be verified that $f'(e_-) = \frac{1-e_-}{(1-e_-)^2 \cdot (1-e_-)} > 0$ and $f'(1-e_-) = \frac{0}{e_-^2 \cdot (1-e_-)} = 0$.

Further differentiate $f'(e_+)$, we get when $e_+ < 1 - ((1-e_-) \cdot e_-^2)^{\frac{1}{3}}$, $f''(e_+) < 0$ and when $e_+ > 1 - ((1-e_-) \cdot e_-^2)^{\frac{1}{3}}$, $f''(e_+) > 0$. Since $e_- < e_+$ and $e_- + e_+ < 1$, we have $e_- < \frac{1}{2}$ and $e_- < 1 - ((1-e_-) \cdot e_-^2)^{\frac{1}{3}} < 1 - e_-$, *i.e.*, $1 - ((1-e_-) \cdot e_-^2)^{\frac{1}{3}}$ locates in the point between $e_-$ and $1 - e_-$. Thus, when $e_- < e_+ < 1 - ((1-e_-) \cdot e_-^2)^{\frac{1}{3}}$, $f(e_+)$ is a strictly concave function and when $1 - ((1-e_-) \cdot e_-^2)^{\frac{1}{3}} < e_+ < 1 - e_-$, $f(e_+)$ is a strictly convex function.

Since $f'(e_-) > 0$ and $f'(1-e_-) = 0$, $e_0$ must locates in the point between $e_-$ and $1 - ((1-e_-) \cdot e_-^2)^{\frac{1}{3}}$ which satisfy $f'(e_0) = 0$. Thus when $e_- < e_+ < e_0$, $f(e_+)$ monotonically increases and when $e_0 < e_+ < 1 - e_-$, $f(e_+)$ monotonically decreases. Since $f(e_-) = f(1-e_-) = 0$. We have $f(e_+) > 0$ when $e_- < e_+ < 1 - e_-$.

Proof done.

We depict a figure in Figure 6 to better show the effect of down-sampling strategy. It can be seen the curves in the figure well support our proposition and proof. When $e_+ - e_-$ is large, down-sampling strategy to make $\mathbb{P}(\widetilde{Y} = 1) = \mathbb{P}(\widetilde{Y} = 0)$ can well decrease the gap even we do not know the true value of $e_-$ and $e_+$. Thus down-sampling strategy can approximate the condition in Theorem 1, which explains why CE with fixed encoder can also work well on instance-dependent label noise in Table 1.

## D  MORE DISCUSSIONS AND EXPERIMENTS

### D.1  WHETHER OR NOT TO FIX THE ENCODER

Nodet et al. (2021) experimentally test many methods and losses based on the pre-trained SSL features and draw the conclusion that the encoder should not be fixed during fine-tuning. We argue that this conclusion is not complete and neglects the specialty of each loss and the gap between SSL and SL varies for each dataset. First, the robustness of CE in Equation (1) is based on the fact that CE is a calibrated and convex loss respect to the linear classifier. However, the losses tested by Nodet et al. (2021) do not all satisfy this property when performing fine-tuning. For example, GCE

Table 2: Comparing CE (fixed encoder) with robust losses on CIFAR10

| Method | Symm | | | |
|---|---|---|---|---|
| | 0.2 | 0.4 | 0.6 | 0.8 |
| SCE | 87.63 | 85.34 | 80.07 | 53.81 |
| APL | 89.22 | 86.05 | 79.78 | 55.06 |
| Peer | 90.70 | 88.29 | 82.10 | 33.03 |
| CE (fixed encoder) | 91.06 | 90.73 | 90.2 | 88.24 |

(Zhang & Sabuncu, 2018) is not convex with respect to linear classifier. Second, if the gap between SSL and SL is large, then the encoder should not be fixed since supervised features can exhibit better performance. However, if the gap between SSL and SL is very small, it is guaranteed by Theorem 1 and Theorem 2 that CE on noisy dataset can approximates SL on clean dataset when encoder is fixed. The observation can be seen in Figure 3 (a). Thus, the above argument suggests that whether to fix the encoder or not is determined by the gap between SSL and SL for a given dataset.

## D.2 PROBLEM OF OTHER ROBUST LOSSES

Many robust losses have been proposed in recent years. For example, some losses (Natarajan et al., 2013; Liu & Guo, 2020) are also proven statistically robust which satisfy:

$$\min_{g,f} \mathbb{E}_{X,Y}[\mathsf{CE}(g(f(X)),Y)] \iff \min_{g,f} \mathbb{E}_{X,\widetilde{Y}}[\ell(g(f(X)),\widetilde{Y})] \qquad (25)$$

where $\ell$ is the robust loss and $f$, $g$ in Equation (25) are simultaneously updated. However, Equation (25) not only needs sufficient training samples to achieve small learning error $\varepsilon$, but also does not consider the optimization difficulty of deep neural networks. Because of the great non-convexity of $f$, it is very hard to guarantee that $f$ and $g$ will converge to global optimal when noise rate varies. Thus the performance of $\ell$ in Equation (25) may suffer severe performance degradation under high noise rate. However, for Equation (1), when $f$ is fixed, CE loss with respect to $g$ is strictly convex which is very friendly for optimization.

To further verify our analysis, we compare CE with SSL pre-trained encoder with some robust losses including SCE (Wang et al., 2019), APL (Ma et al., 2020) and Peer loss (Liu & Guo, 2020) on CIFAR10 with symmetric label noise. The results are shown in Table 2. It can be observed that when noise rate is very high, the performance of these robust losses will drop significantly. However, when SSL pre-trained encoder is fixed, CE loss with respect to linear classifer is strictly convex. Thus the performance is more stable when noise rate increases.

## D.3 THE EFFECT OF DISTANCE MEASURE IN EQUATION (4)

In this paper and experiment, we use $l_2$ norm to calculate the feature distance between SL features and square $l_2$ norm to calculate the distance between SSL features. This choice can lead to good performance from Theory 3 and Figure 5. Practically, since structure regularization mainly captures the relations, different choice does not make a big effect on the performance. We perform an experiment in Figure 7 which shows that the performance of both types are quite close.

## D.4 ABLATION STUDY

In Figure 4, SSL training is to provide SSL features to regularize the output of linear classifier $g$. However, SSL training itself may have a positive effect on DNN. To show the robustness mainly comes from the regularizer rather than SSL training, we perform an ablation study in Figure 8. From the experiments, it is the regularizer that alleviates over-fitting problem of DNN.

## D.5 VISUALIZING FEATURES OF DNN TRAINED BY CE WHEN ENCODER IS NOT FIXED

We visualize the features of DNN trained by CE when encoder is not fixed. Figure 9 shows the Tsne visualization of features before linear classifier under different noise rate. It can be seen, as analyzed

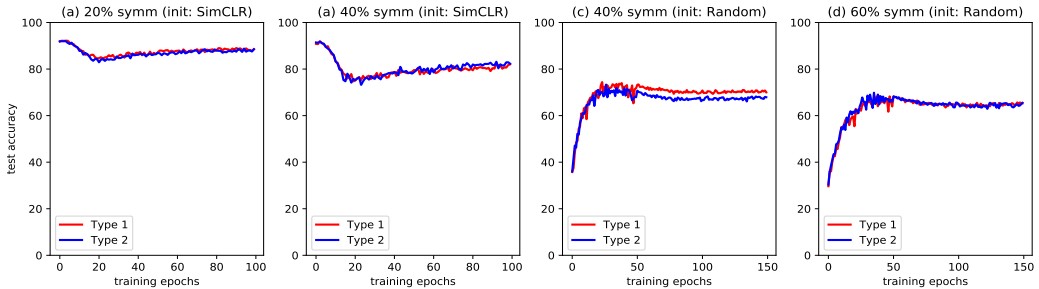

Figure 7: Comparing difference choices of distance measure in Equation (4). Type 1 denotes using $l_2$ norm to calculate distance between SL features and square $l_2$ norm to calculate distance between SSL features, which is adopted in our paper. Type 2 denotes using $l_2$ norm to calculate distance for both SL and SSL features.

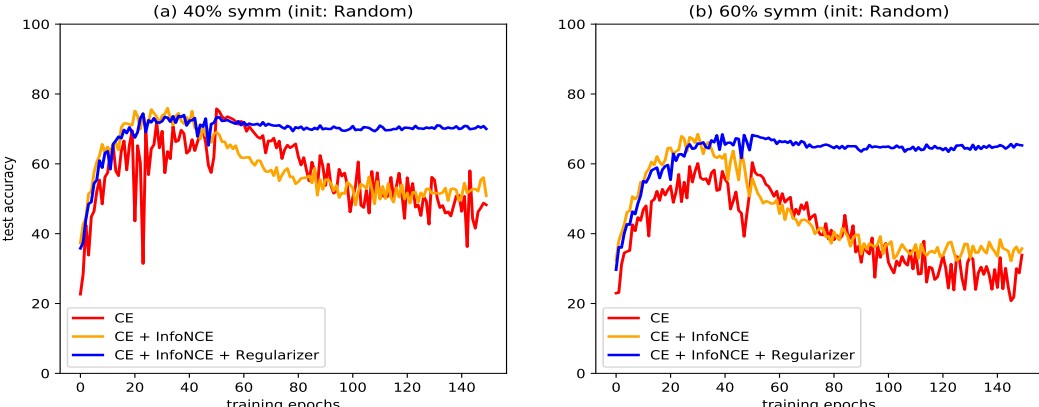

Figure 8: Ablation study of using the regularizer to train DNN on noisy dataset.

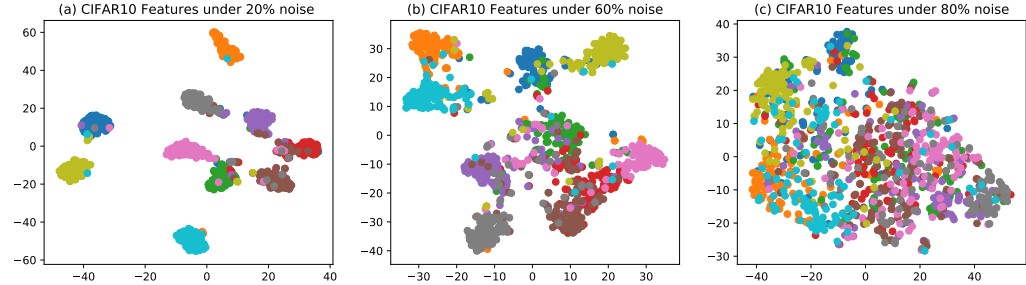

Figure 9: Tsne visualization of supervised features before linear classifier on CIFAR10 under 20%, 60% and 80% symmetric label noise, respectively.

in the paper, due to non-convexity and great over-fitting capability of $f$ (encoder), the features of $f(X)$ have already been contaminated by the label noise before linear classifier, thus CE on vanilla $(X, \widetilde{Y})$ is not robust.

### D.6 THE EFFECT OF DIFFERENT SSL-PRETRAINED METHODS

Our analyses and experiments are not restricted to any specific SSL method. Experimentally, other SSL methods are also adoptable to pre-train SSL encoders. In Figure 3, SimCLR (Chen et al., 2020) is adopted to pre-train SSL encoder. For a comparison, we pre-train a encoder with Moco (He et al., 2020) on CIFAR10 and fine-tune linear classifier on noisy labels in Table 3.

Table 3: Comparing different SSL methods on CIFAR10 with symmetric label noise

| Method | Symm label noise ratio | | | |
|---|---|---|---|---|
| | 0.2 | 0.4 | 0.6 | 0.8 |
| CE (fixed encoder with SimCLR init) | 91.06 | 90.73 | 90.2 | 88.24 |
| CE (fixed encoder with MoCo init) | 91.55 | 91.12 | 90.45 | 88.51 |

It can be observed that different SSL methods have very similar results, which further verifies our Theorems in the paper.

### D.7 EXPERIMENTS ON HUMAN-ANNOTATED NOISY DATASET

We perform experiments on CIFAR100-N (A challenging human-annotated noisy dataset) (Wei et al., 2021) with structure regularization. From Table 4, it can be observed that by using SSL features, Structure Regularization can outperform many benchmark methods on real-world human annotated noisy dataset.

Table 4: The best epoch (clean) test accuracy for each method on CIFAR100-N.

| Method | CE | Forward $T$ | GCE | JoCoR | Peer Loss | ELR | Structure Regularization |
|---|---|---|---|---|---|---|---|
| Acc. | 55.50 | 57.01 | 56.73 | 59.97 | 57.59 | 58.94 | **61.12** |

## E    DETAILED SETTING OF EXPERIMENTS

**Datasets:** We use DogCat, CIFAR10 and CIFAR100 for experiments. DogCat has 25000 images. We randomly choose 24000 images for training and 1000 images for testing. For CIFAR10 and CIFAR100, we follow standard setting that use 50000 images for training and 10000 images for testing.

**Setting in Section 3:** SimCLR is deployed for SSL pre-training with ResNet50 for DogCat and ResNet34 for CIFAR10 and CIFAR100. Each model is pre-trained by 1000 epochs with Adam optimizer (lr = 1e-3) and batch-size is set to be 512. During fine-tuning, we fix the encoder and only fine-tune the linear classifier on noisy dataset with Adam (lr = 1e-3) for 100 epochs and batch-size is set to be 256.

**Setting in Section 4:** The basic hyper-parameters are identical to Section 3, except when we train from scratch (random initialization), learning rate is set to be 0.1 at initial state and is decayed by 0.1 at 50 epochs. In Equation (7), we use MSE loss for measuring the relations between SL features and SSL features. However, since MSE loss may cause gradient exploration when prediction is far from ground-truth, we use smooth $l_1$ loss instead. Smooth $l_1$ loss is an enhanced version of MSE loss. When prediction is not very far from ground-truth, smooth $l_1$ loss is MSE, and MAE when prediction is far.

The code with running guideline has already been attached in the supplementary material.

## F    GENERALIZE THEOREM 1 TO MULTI-CLASS CLASSIFICATION

**Theorem 5** *For K-class classification, let* $g_1 = \arg\min_{g \in \mathcal{G}} \mathbb{E}_{\mathcal{D}}[\mathsf{CE}(g(f(X)), Y)]$, $g_2 = \arg\min_{g \in \mathcal{G}} \mathbb{E}_{\widetilde{\mathcal{D}}}[\mathsf{CE}(g(f(X)), \widetilde{Y})]$. *Then if symmetric noise ratio* $\epsilon < \frac{K-1}{K}$, *for each sample* $x_i$, *we have*

$$\arg\max_k g_1(f(x_i))_k = \arg\max_k g_2(f(x_i))_k \tag{26}$$

*where* $f$ *is fixed encoder and* $g$ *is the linear classifier,* $g(f(\cdot))$ *denotes the network output after softmax layer.*

**proof:** Following the proof for Theorem 1, suppose $g_1^{\text{Bayes}}$ and $g_2^{\text{Bayes}}$ are Bayes optimal classifiers under $\mathbb{P}(f(X), Y)$ and $\mathbb{P}(f(X), \widetilde{Y})$, respectively. We aim to show $g_1^{\text{Bayes}}$ and $g_2^{\text{Bayes}}$ are consistent. *I.e.,* the decision boundaries are the same.

Denote $X_i = X | Y = i$. Define $p_i(\boldsymbol{x})$ to be the distribution of $f(X_i)$. According to the bayes rules:

$$\mathbb{P}(Y = i | \boldsymbol{x}) = \frac{\mathbb{P}(Y = i) \cdot p_i(\boldsymbol{x})}{\sum_{j=1}^{K} P(Y = j) \cdot p_j(\boldsymbol{x})} \tag{27}$$

Let $\mathcal{X}$ be the space of $f(X)$:

$$\mathcal{X} = \mathcal{X}_1 \cup \mathcal{X}_2 \cdots \cup \mathcal{X}_K \tag{28}$$

where $\mathcal{X}_i = \{\boldsymbol{x} \in f(X) | \mathbb{P}(Y = i | \boldsymbol{x}) > \mathbb{P}(Y = j | \boldsymbol{x}), i \neq j\}$.

Let $p_i^{\epsilon}(\boldsymbol{x})$ be the distribution of samples whose observed labels are $i$ under symmetric label noise rate $\epsilon$. We have:

$$p_i^{\epsilon}(\boldsymbol{x}) = m_i \cdot \left\{ \mathbb{P}(Y = i) \cdot (1 - \epsilon) \cdot p_i(\boldsymbol{x}) + \sum_{j \neq i} \mathbb{P}(Y = j) \cdot \frac{\epsilon}{K - 1} \cdot p_j(\boldsymbol{x}) \right\} \tag{29}$$

where $m_i$ is the normalization term to make distribution valid. Specifically, $m_i = \frac{1}{\mathbb{P}(Y=i)\cdot(1-\epsilon)+\sum_{j\neq i}\mathbb{P}(Y=j)\cdot\frac{\epsilon}{K-1}}$. Under label noise, $\mathbb{P}(\widetilde{Y} = i) = \mathbb{P}(Y = i) \cdot (1 - \epsilon) + \sum_{j \neq i} \mathbb{P}(Y = j) \cdot \frac{\epsilon}{K-1} = \frac{1}{m_i}$. According to the bayes rules, we have:

$$\mathbb{P}(\widetilde{Y} = i | \boldsymbol{x}) = \frac{\mathbb{P}(\widetilde{Y} = i) \cdot p_i^{\epsilon}(\boldsymbol{x})}{\sum_{j=1}^{K} P(\widetilde{Y} = j) \cdot p_j^{\epsilon}(\boldsymbol{x})} \tag{30}$$

Let $\mathcal{X} = \mathcal{X}_1^{\epsilon} \cup \mathcal{X}_2^{\epsilon} \cdots \cup \mathcal{X}_K^{\epsilon}$, where $\mathcal{X}_i^{\epsilon} = \{\boldsymbol{x} \in f(X) | \mathbb{P}(\widetilde{Y} = i | \boldsymbol{x}) > \mathbb{P}(\widetilde{Y} = j | \boldsymbol{x}), i \neq j\}$. To prove $g_1^{\text{Bayes}}$ and $g_2^{\text{Bayes}}$ are consistent, it is equivalent to show $\mathcal{X}_i = \mathcal{X}_i^{\epsilon}$ when $\epsilon < \frac{K-1}{K}$.

Without loss of generality, we prove $\mathcal{X}_1 = \mathcal{X}_1^{\epsilon}$ when $\epsilon < \frac{K-1}{K}$.

When $j \neq 1$, we have:

$$
\begin{aligned}
&\mathbb{P}(\widetilde{Y} = 1 | \boldsymbol{x}) - \mathbb{P}(\widetilde{Y} = j | \boldsymbol{x}) \\
&= \frac{(1 - \epsilon - \frac{\epsilon}{K-1}) \cdot \mathbb{P}(Y = 1) \cdot p_1(\boldsymbol{x}) - (1 - \epsilon - \frac{\epsilon}{K-1}) \cdot \mathbb{P}(Y = j) \cdot p_j(\boldsymbol{x})}{\text{normalization}}
\end{aligned} \tag{31}
$$

Since $\mathcal{X}_1 = \{\boldsymbol{x} \in f(X) | \mathbb{P}(Y = 1 | \boldsymbol{x}) > \mathbb{P}(Y = j | \boldsymbol{x}), j \neq 1\}$, $\mathcal{X}_1 = \mathcal{X}_1^{\epsilon}$ if $1 - \epsilon - \frac{\epsilon}{K-1} > 0$, *i.e.,* $\epsilon < \frac{K-1}{K}$.

Proof Done.

# G  ILLUSTRATING BAYES CLASSIFIER IS NOT CONSISTENT UNDER ASYMMETRIC AND INSTANCE LABEL NOISE

**Asymmetric label noise:** We give an example on K-class classification (K=3). Following the definition and notation in Section F, suppose the asymmetric label noise ratio is $\epsilon$, which is generated between two adjacent label ($1 \Leftrightarrow 2, 2 \Leftrightarrow 3, 3 \Leftrightarrow 1$), we have:

$$
\begin{aligned}
&\mathbb{P}(\widetilde{Y} = 1 | \boldsymbol{x}) - \mathbb{P}(\widetilde{Y} = 2 | \boldsymbol{x}) \\
&= \frac{(1 - \epsilon) \cdot \mathbb{P}(Y = 1) \cdot p_1(\boldsymbol{x}) - (2 \cdot \epsilon - 1) \cdot \mathbb{P}(Y = 2) \cdot p_2(\boldsymbol{x}) - \epsilon \cdot \mathbb{P}(Y = 3) \cdot p_3(\boldsymbol{x})}{\text{normalization}}
\end{aligned} \tag{32}
$$

$$\mathbb{P}(\widetilde{Y} = 1|\boldsymbol{x}) - \mathbb{P}(\widetilde{Y} = 3|\boldsymbol{x})$$

$$= \frac{(1 - 2 \cdot \epsilon) \cdot \mathbb{P}(Y = 1) \cdot p_1(\boldsymbol{x}) - \epsilon \cdot \mathbb{P}(Y = 2) \cdot p_2(\boldsymbol{x}) - (1 - \epsilon) \cdot \mathbb{P}(Y = 3) \cdot p_3(\boldsymbol{x})}{\text{normalization}} \quad (33)$$

Since $\mathcal{X}_1 = \{\boldsymbol{x} \in f(X)|\mathbb{P}(Y = 1|\boldsymbol{x}) > \mathbb{P}(Y = j|\boldsymbol{x}), j \neq 1\}$ and $\mathcal{X}_1^\epsilon = \{\boldsymbol{x} \in f(X)|\mathbb{P}(\widetilde{Y} = 1|\boldsymbol{x}) > \mathbb{P}(\widetilde{Y} = j|\boldsymbol{x}), j \neq 1\}$, it is easy to verify $\mathcal{X}_1 = \mathcal{X}_1^\epsilon$ only when $\epsilon = 0$. Thus asymmetric label noise does not induce consistent Bayes classifier. We also provide an experiment on CIFAR10 to further verify our analysis in Table 5. It can be seen that for asymmetric label noise, there is only marginal improvement compared to symmetric label noise in Figure 3.

Table 5: Comparing CE (fixed encoder) with baseline (CE with random initialized unfixed encoder) on CIFAR10 with asymmetric label noise

| Method | Asymm label noise ratio | | | |
|---|---|---|---|---|
| | 0.1 | 0.2 | 0.3 | 0.4 |
| CE (baseline, unfixed encoder with random init) | 90.69 | 88.59 | 86.14 | 80.11 |
| CE (fixed SSL pre-trained encoder) | 90.85 | 89.92 | 88.25 | 82.46 |

**Instance-dependent label noise**: Instance-dependent label noise behaves like symmetric label noise, *i.e.,* the noisy labels may exist in all the classes. However, the noise ratio for each class is not the same and dependent on the features. From Section F, when noise ratio for each class is not the same, the Bayes classifier is not consistent. However, from Table 1 and Section C we theoretically and experimentally show that simple down-sampling strategy can approximate condition in Theorem 1 and Theorem 5 which is helpful for instance level label noise.

