# OpenReview forum: "Demystifying How Self-Supervised Features Improve Training from Noisy Labels"
_ICLR.cc/2022/Conference — ICLR 2022 Submitted_

### Official Review · Reviewer_WCJg · 2021-11-02

**Correctness:** 3
**Technical Novelty And Significance:** 2
**Empirical Novelty And Significance:** 2
**Recommendation:** 5
**Confidence:** 2

**Main Review:**

Strengths:
- Understanding how self-supervised learning can be used, and particular with regards to the impact of fine tuning representations is an important question, as is dealing with label noise.
- The paper has quite a large scope of topics that it tries to connect.

Weaknesses:
- I find the paper somewhat fragmented, jumping between different ideas and topics without going too deep enough into any single one. It's not really clear to me that section 4 is related to section 2/3, they seem to studying quite distinct subjects?
- I find the lack of analysis regarding the specific type of SSL algorithm you use, and the effects that that may have, a weakness. The authors sidestep this by assumptions 1 and 4, but as seen in Figure 2c) on standard small vision datasets these assumptions do not seem to hold (never mind imagenet). What effect does using different SSL methods have on the results, even empirically?
- Many of the results do not seem particularly surprisingly to me and I'm not sure if the application to noisy labels is sufficiently novel. For example, it seems intuitive that one should not fine tune a pretrained NN if the fine tuning dataset is small, i.e. that there isn't much 'signal'. Likewise, it seems intuitive that one should do the same when faced with noisy data.
- It is not clear to me how you train h, and with what objective, in eq 3? If h is the regulariser surely it cant be trained towards g? Please can you explain this to me.
- I think the setting for fig 3 is a bit strange. Surely you should not stop training at the best test accuracy but rather for a fixed number of epochs to show the 'damage' of training with different noise levels for a fixed amount of training. It seems like for higher noise rates that the you are taking very few epochs by stopping based on test accuracy. The results would look more in your favour too I imagine and reflect well on fixed encoder, given fig 3d.

Minor comment:
- Is X_+ for label 1 or 0, it is defined as label 1 but in Theorem 3 it is 0?

**Summary Of The Paper:**

This paper studies the usefulness of self-supervised features when encountering data with noisy labels. It presents an array of ideas on this topic: including the question of fine-tuning pre-trained representations, using ideas from distillation as regularisation to improve generalisation. Some theoretical and empirical results are relating to the authors arguments.

**Summary Of The Review:**

I do not recommend the paper for acceptance currently as I do not think it presents a coherent picture (of a very interesting topic) and also have some confusions about the presentation of the paper.

---

> ### Author Response · Authors · 2021-11-19
> **Response to Reviewer WCJg (part1)**
>
> Thanks for reviewing our paper and providing suggestions. Please see our response below:
>
> **Question 1:** I find the paper somewhat fragmented, jumping between different ideas and topics without going too deep enough into any single one. It's not really clear to me that section 4 is related to section 2/3, they seem to studying quite distinct subjects?
>
> **Response 1:** To the best of our knowledge, we are the first to provide theoretical analyses on why and how SSL features relate to learning with noisy labels. Our theories (Theorem 1, Theorem 3, Theorem 5) explain many (surprising) observations in a line of papers that use SSL features to deal with label noise [R1, R2, R3] and can serve as a guideline for future research.
>
>
>  **Takeaways** in Section 3 explains the relation between Section 3 and Section 4. Section 3 mainly shows the benefits of only fine-tuning the linear classifier  **with fixed SSL pre-trained encoder** and analyzes **when encoders should not be fixed**. Section 4 complements Section 3 and discusses how to use SSL features **when encoder can not be fixed**.
> We will better present our theories and experiments in the final version.
>
>
> [R1] A. Ghosh and A. Lan. Contrastive learning improves model robustness under label noise. In Proceedings of the IEEE/CVF Conference on Computer Vision and Pattern Recognition, pages 2703–2708, 2021.
>
> [R2] P. Nodet, V. Lemaire, A. Bondu, and A. Cornuéjols. Contrastive representations for label noise require fine-tuning. arXiv preprint arXiv:2108.09154, 2021.
>
> [R3] C. Tan, J. Xia, L. Wu, and S. Z. Li. Co-learning: Learning from noisy labels with self-supervision. arXiv preprint arXiv:2108.04063, 2021.
>
> **Question 2:** I find the lack of analysis regarding the specific type of SSL algorithm you use, and the effects that that may have, a weakness. The authors sidestep this by assumptions 1 and 4, but as seen in Figure 2c) on standard small vision datasets these assumptions do not seem to hold (never mind imagenet). What effect does using different SSL methods have on the results, even empirically?
>
> **Response 2:** To avoid confusion, we would like to first highlight that Theorem 1 and Theorem 5 do not rely on any specific distribution of SSL features. Assuming SSL features to be Gaussian (Assumption 1) is mainly to find a tractable analytical model to show that good SSL features can induce classifier with high performance on clean data (Theorem 2), which is generally admitted in the SSL community [R1, R2]. The Gaussian distribution is also assumed in the prediction of pre-trained models [R3]. We also have Figure 2 to help us justify the Gaussian assumption. Figure 2(a) and 2(b) explicitly show there are clear decision boundaries between different classes. By the Central Limit Theorem, it is natural to assume the distribution within each cluster of data to be Gaussian. Although there are some exceptions in Figure 2(c) when there are 100 classes, it may due to the information loss during compressing a 128-dimension vector to this 2-D figure.
> Other reasonable distributions may also be assumed, but it does not change our conclusion in Theorem 1 and Theorem 5.
>
>
> Secondly, Our analyses and experiments are not restricted to any specific SSL method.
> Experimentally, other SSL methods are also adoptable to pre-train SSL encoders.
> In our initial submission, SimCLR [R1] is adopted to pre-train SSL encoder. For a comparison, we pre-train a encoder with Moco
> [R2] on CIFAR10 and fine-tune linear classifier on noisy labels. The results are as follows:
>
> |               Method                | sym. 0.2 | sym. 0.4 | sym. 0.6 | sym. 0.8 |
> | :---------------------------------: | :------: | :------: | :------: | :------: |
> | CE (fixed encoder with SimCLR init) |  91.06   |  90.73   |   90.2   |  88.24   |
> |  CE (fixed encoder with MoCo init)  |  91.55   |  91.12   |  90.45   |  88.51   |
>
>
> It can be observed that different SSL methods have very similar results which both suggest that CE with fixed SSL pre-trained encoder is robust to symmetric label noise.  We have added MoCo experiments in our revised version (Section D.6).
>
>
> [R1] Kaiming He, Haoqi Fan, Yuxin Wu, Saining Xie, and Ross Girshick. Momentum contrast for unsupervised visual representation learning. In Proceedings of the IEEE/CVF Conference on Computer Vision and Pattern Recognition, pp. 9729–9738, 2020.
>
> [R2] Ting Chen, Simon Kornblith, Mohammad Norouzi, and Geoffrey Hinton. A simple framework for contrastive learning of visual representations. In International conference on machine learning, pp. 1597–1607. PMLR, 2020.
>
> [R3] Lee, K., Yun, S., Lee, K., Lee, H., Li, B. and Shin, J., 2019, May. Robust inference via generative classifiers for handling noisy labels. In International Conference on Machine Learning (pp. 3763-3772). PMLR.

---

> > ### Comment · Reviewer_WCJg · 2021-11-23
> > **still not sure about assumption 1**
> >
> > Thanks for taking the time to respond to my review.
> >
> > I'm still confused where SSL comes into assumption 1 at all. Surely you could have assumption 1 (or a similar assumption for non-gaussian features) for any sort of pretraining (supervised or not), so it seems a bit strange that the paper is focused on SSL when there is assumption 1 which means that one can avoid alot of the problems associated to SSL training (like feature collapse) and arrive at a nice setting (of seperated classes) which is crucial for the later theory. It seems to me that the framing of theorem 1 is "good SSL encoders help when training on noisy labels" but in reality it's "When you have a nice condition on the pretrained features then training on noisy labels works" where the latter has no relevance to SSL? Please can the authors clarify this for me.
> >
> > Thank you for adding MoCo experiments, though I would suggest that you retarget your work to "contrastive" SSL as opposed to general SSL, as there are plenty of other non-contrastive SSL methods such as BYOL/SimSiam/Barlow Twins (these are just the most recent ones that I am more familiar with but there are previous SSL works before SimCLR) that it would be good to add experimental evidence for if you would like to say the paper addresses general SSL.
> >
> > Finally, thanks for adding new settings beyond symmetric label noise. I'd like to ask if, under assumption 1 if we do actually have different classifiers from symmetric label noise (in the sense that whp all points will lie far away from the decision boundary, so even if we have different decision boundaries for different noises in practice it will not make a difference under assumption 1).

---

> > > ### Author Response · Authors · 2021-11-24
> > > **Further clarification (part 1)**
> > >
> > > Thank you for your reply. Please see our responses below:
> > >
> > > **Question 1:** I'm still confused where SSL comes into assumption 1 at all. Surely you could have assumption 1 (or a similar assumption for non-gaussian features) for any sort of pretraining (supervised or not), so it seems a bit strange that the paper is focused on SSL when there is assumption 1 which means that one can avoid alot of the problems associated to SSL training (like feature collapse) and arrive at a nice setting (of separated classes) which is crucial for the later theory. It seems to me that the framing of theorem 1 is "good SSL encoders help when training on noisy labels" but in reality it's "When you have a nice condition on the pretrained features then training on noisy labels works" where the latter has no relevance to SSL? Please can the authors clarify this for me.
> > >
> > > **Response 1:**
> > >
> > > - **Explaining Assumption 1:** We want to highlight that **Assumption 1 only serves as a tractable way to find a closed-form solution in Theorem 2, Eq. (2)**. The Gaussian assumption is mainly to show good SSL features can induce classifier with high performance on clean data.
> > >     Note there exists empirical evidences showing that SSL features tend to follow a Gaussian distribution. For example, in [R1], Figure 3(c) shows that, after some transformations, the "angles" of features in Class 0, Class 3, Class 6, and Class 9 approximately follow Gaussian distributions with different means.
> > >     It is also worth noting that, other theorems, such as Theorem 1 and Theorem 5, are proved **without specifying the distribution of SSL features**.
> > >
> > > - **Why must SSL features rather than Supervised Learning (SL) features?** Good question. Supervised learning on the clean dataset would be one of the best choice if we have a clean dataset. But in learning with noisy labels, we do not have the access to the clean data. In contrast, we only have a noisy dataset. If we train a deep neural network on the noisy dataset, the model will memorize noisy labels thus the representations will be corrupted [R2, R3]. To fully avoid memorizing the noisy patterns, self-supervised learning is a good choice. Supervised learning on other clean datasets may be a good choice but it requires additional data.
> > >     Recently, SSL on noisy labels is a popular solution but how and why the SSL features would help is not clear. We hope our paper could contribute to the understanding of using SSL features in learning with noisy labels.
> > >
> > > - **Framing of Theorem 1:** As mentioned before, we do not need strong condition of SSL features for proving Theorem 1 and Theorem 5. The quality of the encoder is determined by the **linear evaluation** on clean data, which is generally adopted in the SSL community (SimCLR, MoCo,BYOL, etc). Thus the framing of Theorem 1 and Theorem 5, in the context of SSL, is **For any SSL method, if it exhibits good performance on clean data via linear evaluation, then it also exhibits the same good performance on noisy data with symmetric label noise**. We do believe Theorem 1 and Theorem 5 are very applicable since it is not limited to any SSL method. This is also the reason why both MoCo and SimCLR behave very well on symmetric label noise (Table 3 in Section D.6) since they all have good performance on clean data via linear evaluation.
> > >
> > > [R1] Wang, T., \& Isola, P. (2020, November). Understanding contrastive representation learning through alignment and uniformity on the hypersphere. In International Conference on Machine Learning (pp. 9929-9939). PMLR.
> > >
> > >  [R2] Han, B., Yao, Q., Liu, T., Niu, G., Tsang, I. W., Kwok, J. T., and Sugiyama, M. A survey of label-noise representation learning: Past, present and future. arXiv preprint arXiv:2011.04406, 2020.
> > >
> > >  [R3] Liu, Y. Understanding Instance-Level Label Noise: Disparate Impacts and Treatments. In Proceedings of the 38th International Conference on Machine Learning, ICML ’21, 2021
> > >
> > > **Question 2:** Thank you for adding MoCo experiments, though I would suggest that you retarget your work to "contrastive" SSL as opposed to general SSL, as there are plenty of other non-contrastive SSL methods such as BYOL/SimSiam/Barlow Twins (these are just the most recent ones that I am more familiar with but there are previous SSL works before SimCLR) that it would be good to add experimental evidence for if you would like to say the paper addresses general SSL.
> > >
> > > **Response 2:** Very thanks for the suggestion. In our paper, the SSL pre-training methods are indeed all contrastive methods. We will clarify this in our revised version and perform experiments on other types of SSL methods in the future.

---

> > > > ### Comment · Reviewer_WCJg · 2021-12-02
> > > > **keeping my score**
> > > >
> > > > Thanks for the clarifications, I am keeping my score.
> > > >
> > > > You write that *"For any SSL method, if it exhibits good performance on clean data via linear evaluation, then it also exhibits the same good performance on noisy data with symmetric label noise"*, but my problem is that you can remove the word "SSL" from your sentence and it still holds, so your results are not specific to SSL nor any of the problems that we face in SSL.

---

> > > > > ### Author Response · Authors · 2021-12-02
> > > > > **Further reply for a possible misunderstanding**
> > > > >
> > > > > Thanks for your feedback. However, there seems to be a misunderstanding. We use the word "SSL" for several reasons:
> > > > >
> > > > > - Our paper is motivated by the current observations that using SSL features can substantially improve the performance of learning with noisy labels [R1, R2, R3]. However, no theoretical analyses have been made to understand these (surprising) observations. Thus we develop theorems and experiments to analyze why and how SSL feature benefits the training from noisy labels. **Using the word "SSL" is because of our motivation in the first place**.
> > > > >
> > > > > - It is true that our theorems can also be applied to
> > > > > Supervised learning (SL) features. *i.e.,* like the reviewer said, one can remove the "SSL" word from the sentences and it still holds. However, we have explained in our previous response that in learning with noisy labels, we do not have the access to clean data. In contrast, we only have a noisy dataset. If we train a deep neural network on the noisy dataset, the model will memorize noisy labels thus the representations will be corrupted which leads to poor performance [R4, R5]. In other words, **without additional clean data, the SL features will vary for different noise rates (due to memorizing effect of DNN), and Theorem 1 and Theorem 5 can not hold under linear evaluation**.
> > > > >
> > > > > - SSL is indeed a very challenging topic in the literature and many problems are worth being further studied. However, this is out of the scope of our paper. **Our focus is given SSL features, why and how these features can improve the training from noisy labels**.
> > > > >
> > > > > [R1] A. Ghosh and A. Lan. Contrastive learning improves model robustness under label noise. In Proceedings of the IEEE/CVF Conference on Computer Vision and Pattern Recognition, pages 2703–2708, 2021.
> > > > >
> > > > > [R2] P. Nodet, V. Lemaire, A. Bondu, and A. Cornuéjols. Contrastive representations for label noise require fine-tuning. arXiv preprint arXiv:2108.09154, 2021.
> > > > >
> > > > > [R3] C. Tan, J. Xia, L. Wu, and S. Z. Li. Co-learning: Learning from noisy labels with self-supervision. arXiv preprint arXiv:2108.04063, 2021.
> > > > >
> > > > >
> > > > >  [R4] Han, B., Yao, Q., Liu, T., Niu, G., Tsang, I. W., Kwok, J. T., and Sugiyama, M. A survey of label-noise representation learning: Past, present and future. arXiv preprint arXiv:2011.04406, 2020.
> > > > >
> > > > > [R5] Liu, Y. Understanding Instance-Level Label Noise: Disparate Impacts and Treatments. In Proceedings of the 38th International Conference on Machine Learning, ICML ’21, 2021

---

> > > > > > ### Comment · Reviewer_WCJg · 2021-12-02
> > > > > > **discussion**
> > > > > >
> > > > > > Thanks for the response, I am still not convinced sadly. I think you should justify more assumption 1, perhaps showing when it holds and when it is applicable for SSL and not standard supervised learning. You say that *"If we train a deep neural network on the noisy dataset, the model will memorize noisy labels thus the representations will be corrupted which leads to poor performance [R4, R5]"*, but then I would try to show an explicit example where this is the case for SL and that SSL is needed to obtain good representations (ie assumption 1). There are valid regimes of deep NN training when no feature learning occurs and hence the representations will be same as the representations at initialisation, e.g. [1]. Assumption 1 can hold in those cases depending on structures of the inputs and NN architecture, in which case SL with noisy labels will still work.
> > > > > >
> > > > > > [1] https://arxiv.org/abs/1902.06720

---

> > > > > > > ### Author Response · Authors · 2021-12-02
> > > > > > > **Re: discussion**
> > > > > > >
> > > > > > > Thanks for your very fast reply. Please see our response below:
> > > > > > >
> > > > > > > - We still want to emphasize again that Assumption 1 only serves as a tractable way to find a closed-form solution in Eq. (2), showing SSL feature has good performance on clean data via linear evaluation, which is **NOT** related to our main contribution in Section 3. Our main contribution in Section 3 is to discuss the benefits of the fixed encoder and the condition of achieving the consistent classifier with fixed encoder under noisy labels (Theorem 1, Theorem 5). Theorem 1 and Theorem 5 are proved without Assumption 1.
> > > > > > >
> > > > > > >
> > > > > > > - SL with noisy labels is hard to be guaranteed to avoid memorizing noisy patterns. There are many empirical evidences in the literature. For example, in Table 3 from [R1], where the SL feature trained with noisy dataset (e.g., Inst 0.4) suffers performance drop compared to those from the clean dataset. To the best of our knowledge, there is no evidence in the literature showing SL with noisy labels can be guaranteed to not memorize noisy labels. Note SSL feature could be independent of noisy labels since there's no label information in SSL.
> > > > > > >
> > > > > > >
> > > > > > > - The paper (https://arxiv.org/abs/1902.06720) pointed by the reviewer is interesting. However, we are not clear how it relates to the SL features with noisy labels. There is no discussion in this paper that how noisy labels affect the network representation.
> > > > > > > Besides, we think this paper does not state that the final representation is the same as the representation at initialization. This paper is mainly to show for wide neural networks, in the infinite width limit, they are governed by a linear model obtained from the first-order Taylor expansion of the network around its initial parameters.
> > > > > > >
> > > > > > > [R1] Clusterability as an Alternative to Anchor Points When Learning with Noisy Labels. ICML 2021

---

> > > > > > > > ### Comment · Reviewer_WCJg · 2021-12-02
> > > > > > > > **discussion**
> > > > > > > >
> > > > > > > > Thanks for the response, though I think we may be going in circles here and my decision has not be swayed by the author responses thus far, so I may not respond unless felt compelled by the subsequent author response.
> > > > > > > >
> > > > > > > > - Yes but assumption 1 (not the gaussianity, but rather the separation between classes) is crucial for theorem 2, and there is no provable 'benefit of the fixed encoder' if there is no separation between the classes (in terms of $\mu_1,\mu_2,\sigma$), i.e. if $\mu_1=\mu_2$ then theorem 2 just gives risk of 1/2, which is not useful.
> > > > > > > >
> > > > > > > > - Ok perhaps I didn't need to use NTK as an example, instead consider simply initialising a wide NN but not training any of the hidden parameters and only the last layer instead. Then the representation will be completely independent of the noisy labels by definition and identical to the representation at init. This is related to the infinite width NNGP literature which treats the NN as a data-independent kernel/GP [1-2] and is a valid training regime for NNs, which can perform competitively on standard benchmarks [3-4]. Here my point stands in that if the NNGP kernel and data are well suited to each other then assumption 1 can hold and the NN will perform well with noisy labels and so your theory is not able to show any of the benefits of SSL relative to SL.
> > > > > > > >
> > > > > > > > [1] https://arxiv.org/abs/1804.11271
> > > > > > > > [2] https://arxiv.org/abs/1711.00165
> > > > > > > > [3] https://arxiv.org/abs/2003.02237
> > > > > > > > [4] https://arxiv.org/abs/2007.15801

---

> > > > > > > > > ### Author Response · Authors · 2021-12-02
> > > > > > > > > **discussion**
> > > > > > > > >
> > > > > > > > > Dear Reviewer WCJg,
> > > > > > > > >
> > > > > > > > > We did not discuss how an NN with infinite width performs under the noisy label settings in our paper. We would like to highlight several points as follows.
> > > > > > > > >
> > > > > > > > > *  "Simply initializing a wide NN but not training any of the hidden parameters and only the last layer instead"  -Yes, in this case, the representations are independent of label noise. But it cannot show "initializing a wide NN with SSL" is not beneficial.
> > > > > > > > >
> > > > > > > > > * The provided references did not discuss the label noise settings. It is not clear how such architecture and your considered scenario perform in the settings with label noise.
> > > > > > > > >
> > > > > > > > > * It is also not clear why we must consider a wide NN in our paper.
> > > > > > > > >
> > > > > > > > > * We would like to invite the reviewer to think about one simple question: Given one noisy dataset, could we guarantee that SL with a noisy dataset **does not memorize** noisy patterns (in the representation level) when we learn with a deep NN? If yes, we would appreciate your explanations. If not, SSL is one tractable approach that could be beneficial since it is at least **independent of label noise** and should be **better than random initialization**.
> > > > > > > > >
> > > > > > > > > Thanks for your time in this discussion!
> > > > > > > > >
> > > > > > > > > Best,
> > > > > > > > >
> > > > > > > > > ICLR 2022 Conference Paper675 Authors

---

> > > > > > > > > > ### Comment · Reviewer_WCJg · 2021-12-02
> > > > > > > > > > **answer to question**
> > > > > > > > > >
> > > > > > > > > > Yes, the NTK regime will do, as NTK is a data-independent kernel. But also we obviously need an assumption equivalent to assumption 1 (just like you have used for theorem 2), and this time it will stipulate conditions on the NTK (e.g. within-class NTK values are larger than out-of-class).
> > > > > > > > > >
> > > > > > > > > > I am not saying you need to consider wide NNs, but I am using wide NNs to show that your current arguments and assumptions are insufficient to justify why your setting is specific to SSL and why SSL is beneficial for noisy labels relative to vanilla SL.

---

> > > ### Author Response · Authors · 2021-11-24
> > > **Further clarification (part 2)**
> > >
> > > **Question 3:** Finally, thanks for adding new settings beyond symmetric label noise. I'd like to ask if, under assumption 1 if we do actually have different classifiers from symmetric label noise (in the sense that whp all points will lie far away from the decision boundary, so even if we have different decision boundaries for different noises in practice it will not make a difference under assumption 1).
> > >
> > > **Response 3:**  As mentioned before, the classifier is consistent for symmetric label noise, *i.e.,* the decision boundary does not change **with or without Assumption 1**. We think the reviewer wants to ask that in the sense that all points lie far away from the decision boundary, if it will not make a difference when decision boundary changes?
> > >
> > > The intuition is true. If all points are far away from the decision boundary. The small change on the decision boundary will not make a difference on test performance. However,  this condition can not hold in practice. For example, in the image classification tasks, many classes have overlap semantics especially for the dataset with more categories. Thus the features of these classes may be very close to the decision boundary. Figure 2 and Figure 9 in the paper show SSL and SL features via T-SNE. It can be clearly observed that the overlap between features of classes. Further, In Section G, we have proved that decision boundary changes under asymmetric label noise
> > > and the corresponding experiment (Table 5) also shows that when decision boundary changes, the performance of classifier drops significantly under high noise rate.

---

> ### Author Response · Authors · 2021-11-19
> **Response to Reviewer WCJg (part2)**
>
> **Question 3:**  Many of the results do not seem particularly surprisingly to me and I'm not sure if the application to noisy labels is sufficiently novel. For example, it seems intuitive that one should not fine tune a pretrained NN if the fine tuning dataset is small, i.e. that there isn't much 'signal'. Likewise, it seems intuitive that one should do the same when faced with noisy data.
>
> **Response 3:**
>
> - The intuition is true: If there isn't much "signal", we should not fine-tune a pre-trained NN. We did not derive any counter-intuitive results. In tradition SSL, the data for fine-tuning is clean. Thus less data means less signal. In label noise, higher noise rates indicate less true signal and more misleading signal. From the aspect of information, higher noise rates include less useful information. Thus the high-level intuition of analyses for label noise should be consistent with SSL. More analyses are in Section 3.2.1.
>
> - Traditional SSL setting is very different from the label noise setting since we do not know which part of the signal is useful.
>
> - Besides, fine-tuning linear classifier on noisy data is not as intuitive as for clean data.
>     Different types of label noise have very different outcomes. For example, the classifier trained on noisy data is consistent with the classifier trained on clean data **only for symmetric label noise** (See our response for Reviewer GhTK).
>
> **Question 4:** It is not clear to me how you train h, and with what objective, in eq 3? If h is the regulariser surely it cant be trained towards g? Please can you explain this to me.
>
> **Response 4:** Sorry for the confusion. $h$ is trained by InfoNCE. In eq3, it should be
> $\text{InfoNCE}(h(f(X))$ instead
> of $\text{InfoNCE}(X)$. We have revised this part.
>
> **Question 5:** I think the setting for fig 3 is a bit strange. Surely you should not stop training at the best test accuracy but rather for a fixed number of epochs to show the 'damage' of training with different noise levels for a fixed amount of training. It seems like for higher noise rates that the you are taking very few epochs by stopping based on test accuracy. The results would look more in your favour too I imagine and reflect well on fixed encoder, given fig 3d.
>
> **Response 5:**
>
> - Firstly, we report the best epoch accuracy to show the best generalization ability of each method. It also helps us reduce the chances of selecting a non-optimal model that leads to unfair comparison. Admittedly, a validation dataset can be used to select models for CE or CORES (with unfixed f) in Figure 3 and Table 1. However, it may even make the gap larger since a fixed encoder does not suffer much performance drop in the later training stages as shown in Figure 3(d).
> It is also worth noting that the performance gap between the last epoch and the best epoch is not large when training with the regularizer as Figure 5.
>
> - Secondly,
> in the literature of learning with noisy labels, most of works do not separate a validation set from the training set and only report the best epoch test accuracy in CIFAR [R1, R2, R3]. A few works report both best epoch test accuracy and last epoch test accuracy [R4].
> In Figure 3 (a)(b)(c) and Table 1, the best epoch accuracy is reported. While in Figure 3 (d), both best epoch accuracy and last epoch accuracy can be viewed from the curves. These experiments are mainly to show the benefits of the fixed encoder. We will add more details about experimental setting in the final version.
>
> [R1] Robust Curriculum Learning: from clean label detection to noisy label self-correction, ICLR 2021
>
> [R2] Learning with Feature-Dependent Label Noise: A Progressive Approach, ICLR 2021
>
> [R3] When Optimizing f-Divergence is Robust with Label Noise, ICLR 2021
>
> [R4] DivideMix: Learning with Noisy Labels as Semi-supervised Learning. ICLR 2020.
>
> **Question 6:** Is $X_+$ for label 1 or 0, it is defined as label 1 but in Theorem 3 it is 0?
>
> **Response 6:** As defined in Section 2, $ X_+ = X|Y=1$, which denotes the samples whose true labels are $1$. $ X_- = X|Y=0$, which denotes the samples whose true labels are $0$.
>
> In Theorem 3, it is $X_+^{\text{noisy}}$ rather than
> $X_+$. $X_+^{\text{noisy}}$ is defined as
> $X_+^{\text{noisy}} := X|(\widetilde{Y}=1,Y=0)$, which denotes the  samples whose observed labels are $1$ but true labels are $0$. It can be observed that
> $X_+^{\text{noisy}}$ is a subset of $X_-$.

---

### Official Review · Reviewer_PASS · 2021-11-02

**Correctness:** 3
**Technical Novelty And Significance:** 3
**Empirical Novelty And Significance:** 2
**Recommendation:** 5
**Confidence:** 4

**Main Review:**

Strength:
1. Using SSL representation can help learning with noisy labels is an interesting discovery. Using theirs theories, the authors also motivate why fixed encoders are important
2. The authors motivate a regularizer from knowledge distillation to further improve the training performance.

Weakness:

Theoretically, (1-2); Experimentally (3-6); Clarity(7)
1. The novelty of the theorem 1 and 2 is limited. It is trivially true that when you have majority labels that are not noisy, you can learn an optimal classifier.
2. The authors need to explain or cite when SSL give you representations that satisfy assumption 1.
3. It's not clear whether the authors use a validation set. Validation set is important for model selection in machine learning. The authors seem to choose the best test performance directly.
4. The motivation of downsampling is unclear. The authors mentions that downsampling can help satisfy the condition in Theorem 1. However, with downsampling, the target distribution may change and the Bayes classifier may be different. This may change the conclusion in theorem 1. Does downsampling also help for the baseline CORES?
5. Though the authors claim that they do not aim to get SOTA results, more datasets should be introduced. At least, I would like to see the results on cifar 100.
6. Somehow the authors mention the regularizer in section 4. but do not use it in Table 1. Why is that?
7. The paper can be re-organized. The authors describe the use of SSL first and then describe some experiments. After that, the authors describe the regularizer and some other experiments. They look like two separate parts. I suggest the authors put all the experiments and discussions in one section.

**Summary Of The Paper:**

The authors illustrate why self-supervised learning can help learning in training with noisy labels.
The main contributions are:
1. The authors illustrate theoretically why learning good representation can help learning with noisy labels
2. The authors describe why fixed encoders are important.
3. The authors motivate a regularizer between the SL and SSL.

**Summary Of The Review:**

The authors propose an interesting idea that SSL can help learning with noisy-labels but due to the theoretical, experimental and clarity concerns above, I think the paper is marginally below the acceptance threshold.

---

> ### Author Response · Authors · 2021-11-19
> **Response to Reviewer PASS (part1)**
>
> Thanks for reviewing our paper and providing suggestions. Please see our response below:
>
> **Question 1:** The novelty of the theorem 1 and 2 is limited. It is trivially true that when you have majority labels that are not noisy, you can learn an optimal classifier.
>
> **Response 1:**
>
> We would like to respectfully correct the misunderstanding on the novelty of Theorem 1 and Theorem 2. There are mainly two reasons:
>
> * Firstly, with sufficient large model capacity (deep neural networks) and infinite data, we can learn the noisy data distribution of each individual data point. In this case, simply reporting the class with the maximal model prediction is optimal, *i.e.,* when we have majority labels that are not noisy, the Bayes optimal classifier can be learned. However, when the model capacity is limited, we may not be able to learn the noisy data distribution of all data points. Note Theorem 1 focuses on the simple linear classifier with limited capacity, it cannot be guaranteed that we can learn the noisy data distribution for all data points, thus getting the optimal classifier is not trivial.
> * Secondly, our main purpose is to show under what condition that the classifier trained on noisy data is consistent with the classifier trained on clean data, *i.e.,* the decision boundaries are the same.  From Theorem 1 and Theorem 5 (the extension of Theorem 1 which we newly added in Section F), the classifier is consistent under **symmetric label noise with equal noise ratio for each class**. However, for other noise settings such as asymmetric and instance-dependent label noise, even we have majority labels that are not noisy, the classifier is still not consistent (See our response to Reviewer GhTK).
>
> **Question 2:** The authors need to explain or cite when SSL give you representations that satisfy assumption 1.
>
> **Response 2:** To avoid confusion, we would like to first highlight that Theorem 1 and Theorem 5 do not rely on any specific distribution of SSL features. Assuming SSL features to be Gaussian (Assumption 1) is mainly to find a tractable analytical model to show that good SSL features can induce classifier with high performance on clean data (Theorem 2), which is generally admitted in the SSL community [R1, R2]. The Gaussian distribution is also assumed in the prediction of pre-trained models [R3]. We also have Figure 2 to help us justify the Gaussian assumption. Figure 2(a) and 2(b) explicitly show there are clear decision boundaries between different classes. By the Central Limit Theorem, it is natural to assume the distribution within each cluster of data to be Gaussian. Although there are some exceptions in Figure 2(c) when there are 100 classes, it may due to the information loss during compressing a 128-dimension vector to this 2-D figure.
> Other reasonable distributions may also be assumed, but it does not change our conclusion in Theorem 1 and Theorem 5.
>
> [R1] Kaiming He, Haoqi Fan, Yuxin Wu, Saining Xie, and Ross Girshick. Momentum contrast for unsupervised visual representation learning. In Proceedings of the IEEE/CVF Conference on Computer Vision and Pattern Recognition, pp. 9729–9738, 2020.
>
> [R2] Ting Chen, Simon Kornblith, Mohammad Norouzi, and Geoffrey Hinton. A simple framework for contrastive learning of visual representations. In International conference on machine learning, pp. 1597–1607. PMLR, 2020.
>
> [R3] Lee, K., Yun, S., Lee, K., Lee, H., Li, B. and Shin, J., 2019, May. Robust inference via generative classifiers for handling noisy labels. In International Conference on Machine Learning (pp. 3763-3772). PMLR.

---

> > ### Comment · Reviewer_PASS · 2021-11-20
> > **Are you sure central limit theorem tells you that the distribution with each cluster of data to be Gaussian?**
> >
> > Could you add more reference to the question in the title?
> >
> > I think CLT only tell you the probability distribution of the AVERAGE will closely approximate a normal distribution.

---

> > > ### Author Response · Authors · 2021-11-21
> > > **Explaining Gaussian Assumption**
> > >
> > > Thank you for your reply. We further explain our Gaussian assumption as follows:
> > >
> > > - **Explaining CLT:**  The pre-trained networks in the paper are all CNN-based network structure. From the observation in [R1, R2], the final representation of each class in CNN can be viewed as a linear combination  of many decomposed components such as scenes, textures, materials, colors, *etc.* and different part (filter, layer) of the CNN contributes to different component of representation, a.k.a. disentangled representations [R4, R5]. These observations exist in both supervised and unsupervised (self-supervised) tasks [R3].
> > >     In our first response, we were trying to hint that one can treat each disentangled representation as an independent random variable  with finite mean and variance. Note the disentangled representations for different classes or sub-populations should have different means according to the high-level intuition of SSL (and similar mean for similar sub-populations).
> > >     From **Lyapunov CLT**, the average (or weighted averaging) of these components, *i.e.,* the final representation, follows Gaussian distribution. We did not theoretically prove the feasibility of this assumption. But we hope our explanation helps justify the intuition of this assumption.
> > >
> > > - **Empirical evidence:** Apart from Figure 2 in our paper, other works also have similar observations on SSL features. For example, from Figure 3 in [R6], the SSL features of each class approximately follow Gaussian distribution.
> > >
> > > - We want to emphasize again that Theorem 1 and Theorem 5 do not rely on any specific distribution of SSL features.  Assuming SSL features to be Gaussian is mainly to find a tractable analytical model to show that good SSL features can induce classifier with high performance on clean data.
> > >
> > >
> > > Thanks again for the comments. We will further polish our paper accordingly. Please feel free to let us know if there still exists any confusion.
> > >
> > >
> > >
> > > [R1] Bau, D., Zhou, B., Khosla, A., Oliva, A., \& Torralba, A. (2017). Network dissection: Quantifying interpretability of deep visual representations. In Proceedings of the IEEE conference on computer vision and pattern recognition (pp. 6541-6549).
> > >
> > > [R2] Bau, D., Zhu, J. Y., Strobelt, H., Zhou, B., Tenenbaum, J. B., Freeman, W. T., \& Torralba, A. (2018). Gan dissection: Visualizing and understanding generative adversarial networks. arXiv preprint arXiv:1811.10597.
> > >
> > > [R3] http://netdissect.csail.mit.edu/
> > >
> > > [R4] Wang, T., Yue, Z., Huang, J., Sun, Q., \& Zhang, H. Self-Supervised Learning Disentangled Group Representation as Feature. NeurIPS 2021.
> > >
> > > [R5] Higgins, I., Amos, D., Pfau, D., Racaniere, S., Matthey, L., Rezende, D. and Lerchner, A., 2018. Towards a definition of disentangled representations. arXiv preprint arXiv:1812.02230.
> > >
> > > [R6] Wang, T., \& Isola, P. (2020, November). Understanding contrastive representation learning through alignment and uniformity on the hypersphere. In International Conference on Machine Learning (pp. 9929-9939). PMLR.

---

> > > > ### Comment · Reviewer_PASS · 2021-11-21
> > > > **Futher questions**
> > > >
> > > > You never mention disentangled representations in the paper. It's unclear whether it's your original motivation for assumption 1. I am not sure whether self-supervised learning gives you disentangled representations. Your citation [R4] seems to mention self-supervised learning does not give you fully disentangled representations. And they propose some new algorithms to guarantee the disentangled representations. Do you use their method?
> > > >
> > > > It's hard to believe the final representation is an average of the disentangled representations. The weights may be very different. You may need rigorous assumptions for the weights to satisfy CLT.
> > > >
> > > > For your empirical evidence, can you do a statistical test to see whether they follow Gaussian distributions?

---

> > > > > ### Author Response · Authors · 2021-11-22
> > > > > **Further explanation**
> > > > >
> > > > > We do not elaborately explain the Gaussian assumption in our initial submission because this assumption is only to show good SSL features can induce classifier with high performance on clean data. Empirically, this is generally verified and admitted in the SSL community, *i.e.,* if one have a good SSL pre-trained model, it exhibits great performance even for linear evaluation. Besides, the Gaussian assumption is only one tractable way to find a clean closed-form for Eq.(2) in Theorem 2. Note Theorem 1 and Theorem 5 are proved **without specifying the distribution of SSL features**. The main contribution in Section 3 is that we discuss under what condition the classifier trained on clean data is consistent with the classifier trained on noisy data with fixed encoder (Theorem 1, Theorem 5), and what can we do when classifier is not consistent.
> > > > >
> > > > > [R4] is open publicly after the submission deadline. Thus we can not use their methods for SSL pre-training. We mention [R4] because it also shows SSL features can be viewd as disentangled representations and they propose new algorithm to make the representation more disentangled. Note there are more empirical proof in [R1, R2, R3] that show SSL (and SL) features can be disentangled into many parts.
> > > > >
> > > > > **Weighted Average:** Lyapunov CLT: $S_n = \frac{1}{N}\sum_{n=1}^N X_n$, where $\mu_n = X_n$, $\sigma_n^2 = \text{var}(X_n)$ are finite. For the weighted average case $S_n = \sum_{n=1}^N w_n \cdot X_n, \sum_{n=1}^N w_n = 1$, suppose the weights are $w_1,\cdots,w_N$. Then the weighted average is equivalent to $S_n = \frac{1}{N}\sum_{n=1}^N  (N \cdot w_n) \cdot X_n$. We can simply treat $(N \cdot w_n) \cdot X_n$ as a new random variable with finite mean and variance.
> > > > >
> > > > > It is impractical to do **direct statistical test** to see whether SSL features follow Gaussian distributions because the dimension is very high-dimensional (e.g., 128 dimensions) and the number of data is limited. For example, going trough a binary-valued 128-dimension vector requires at least $2^{128}$ points. Accurately estimating the distribution requires much more data. Decreasing the dimension of SSL features is also non-trivial since it may lose much information. However, we believe that other empirical study in [R6] has convincing conclusion that SSL features of each class approximately follow Gaussian distribution. Please note in [R6], Figure 3(c) shows that, after some transformations, the "angles" of features in Class 0, Class 3, Class 6, and Class 9 approximately follow Gaussian distributions with different means.
> > > > >
> > > > >
> > > > > [R1] Bau, D., Zhou, B., Khosla, A., Oliva, A., \& Torralba, A. (2017). Network dissection: Quantifying interpretability of deep visual representations. In Proceedings of the IEEE conference on computer vision and pattern recognition (pp. 6541-6549).
> > > > >
> > > > > [R2] Bau, D., Zhu, J. Y., Strobelt, H., Zhou, B., Tenenbaum, J. B., Freeman, W. T., \& Torralba, A. (2018). Gan dissection: Visualizing and understanding generative adversarial networks. arXiv preprint arXiv:1811.10597.
> > > > >
> > > > > [R3] http://netdissect.csail.mit.edu/
> > > > >
> > > > > [R4] Wang, T., Yue, Z., Huang, J., Sun, Q., \& Zhang, H. Self-Supervised Learning Disentangled Group Representation as Feature. NeurIPS 2021.
> > > > >
> > > > > [R5] Higgins, I., Amos, D., Pfau, D., Racaniere, S., Matthey, L., Rezende, D. and Lerchner, A., 2018. Towards a definition of disentangled representations. arXiv preprint arXiv:1812.02230.
> > > > >
> > > > > [R6] Wang, T., \& Isola, P. (2020, November). Understanding contrastive representation learning through alignment and uniformity on the hypersphere. In International Conference on Machine Learning (pp. 9929-9939). PMLR.

---

> ### Author Response · Authors · 2021-11-19
> **Response to Reviewer PASS (part2)**
>
> **Question 3:** It's not clear whether the authors use a validation set. Validation set is important for model selection in machine learning. The authors seem to choose the best test performance directly.
>
> **Response 3:**
> Firstly, we report the best epoch accuracy to show the best generalization ability of each method. It also helps us reduce the chances of selecting a non-optimal model that leads to unfair comparison. For example, In Table 1, using a validation dataset to select models for CORES or CE with unfixed $f$ may even make the gap larger since a fixed encoder does not suffer much performance drop in the later training stages as shown in Figure 3(d).
> It is also worth noting that the performance gap between the last epoch and the best epoch is not large when training with the regularizer as Figure 5.
>
> Secondly,
> in the literature of learning with noisy labels, most of works do not separate a validation set from training set and only report the best epoch test accuracy in CIFAR [R1, R2, R3]. A few works report both best epoch test accuracy and last epoch test accuracy [R4].
> In Figure 3 (a)(b)(c) and Table 1, the best epoch accuracy is reported. While in Figure 3(d) and Figure 5, both best epoch accuracy and last epoch accuracy can be viewed from the curves. These experiments, following conventional evaluation criteria, show the benefits of only fine-tuning the linear classifier.
>
> [R1] Robust Curriculum Learning: from clean label detection to noisy label self-correction, ICLR 2021
>
> [R2] Learning with Feature-Dependent Label Noise: A Progressive Approach, ICLR 2021
>
> [R3] When Optimizing f-Divergence is Robust with Label Noise, ICLR 2021
>
> [R4] DivideMix: Learning with Noisy Labels as Semi-supervised Learning. ICLR 2020.
>
>
> **Qustion 4:**  The motivation of downsampling is unclear. The authors mentions that downsampling can help satisfy the condition in Theorem 1. However, with downsampling, the target distribution may change and the Bayes classifier may be different. This may change the conclusion in theorem 1. Does downsampling also help for the baseline CORES?
>
> **Response 4:** We acknowledge that, under instance-dependent label noise, it is hard to guarantee down-sampling can theoretically make the classifier be consistent with the classifier trained on clean data. It may also be true that with down-sampling, the clean label distribution also changes (for example, changing label distribution from balance to imbalance).
> However, Theorem 1 and Theorem 5 are proved without
> specifying the distribution of clean labels, suggesting that
> the classifier is consistent even for imbalanced datasets under symmetric label noise.
> Secondly, with SSL features, classifiers are robust to datasets with imbalanced clean labels [R1, R2].
> Thus down-sampling strategy
> is helpful for instance-dependent label noise because it reduces error rate imbalances to approximate the condition in Theorem 1 and Theorem 5 (Illustration in Section C). We have added these new analyses in the revised version (contents highlighted blue in Section 3.2.2).
>
> We also perform CORES with down-sampling on CIFAR10 with instance dependent label noise (thanks for suggestion). The results are as follows:
>
> |        Method         | ins. 0.2 | ins. 0.4 | ins. 0.6 |
> | :-------------------: | :------: | :------: | :------: |
> |         CORES         |  89.50   |  82.84   |  79.66   |
> | CORES (down-sampling) |  89.15   |  81.62   |  77.67   |
>
>
> It can be seen that down-sampling can not improve CORES for instance-dependent label noise while it can improve the classifier with fixed SSL encoder from our experiments and analyses.
>
> [R1]: Rethinking the Value of Labels for Improving Class-Imbalanced Learning, NeurlPS 2020.
>
> [R2]: Self-supervised Learning is More Robust to Dataset Imbalance, arxiv, 2110.05025
>
>
> **Question 5:** Though the authors claim that they do not aim to get SOTA results, more datasets should be introduced. At least, I would like to see the results on cifar 100.
>
> **Response 5:** We would like to clarify that **we have tested on CIFAR100 in Figure 3(c) and Table 1 *(left panel Inst, CIFAR100)* in our initial submission** to verify our theoretical claims. To test the performance of structure regularization by using SSL features, we also perform experiments on CIFAR100-N (A challenging human-annotated noisy dataset) [R1] as follows:
>
> | Method | CE    | Forawrd-T | GCE   | JoCoR | Peer Loss | ELR   | Structure Regularization |
> | ------ | ----- | :-------: | ----- | ----- | --------- | ----- | :----------------------: |
> | Acc    | 55.50 |   57.01   | 56.73 | 59.97 | 57.59     | 58.94 |          61.12           |
>
>
> It can be seen that by using SSL features, Structure Regularization can outperform many benchmark methods on the real-world human-annotated noisy dataset.
>
> [R1] http://noisylabels.com

---

> > ### Comment · Reviewer_PASS · 2021-11-21
> > **Not using a validation set is not a good way for doing ML research**
> >
> > I am not sure why the authors believe not using a validation set is a good way to do machine learning research. What's the meaning of reporting the best test accuracy without validation? How do you select the best test acc in practice once you do not have the labels. I am not also sure why citing other papers that incorrectly not using validation sets is a good sign of not using it. I can also cite many other papers that use validation set. For example, ResNet paper use a validation set.

---

> > > ### Author Response · Authors · 2021-11-21
> > > **a noisy validation can be tricky and might lead to unfair comparisons**
> > >
> > > Our previous response might not be clear. We didn't intend to say that validation is not preferred here. Rather, if there is a way with consensus to fairly perform model selection on noisy validation data, we'd very much prefer to do so. The problem, however, is that performing validation on a noisy validation set is a non-trivial task. Directly computing model performance using a noisy validation dataset makes little sense. This is a unique challenge that was not encountered in other contexts when ground truth supervision is available, for instance in Resnet as cited by the reviewer.
> > >
> > > Previous results have promoted each of their own and different ways of validating using noisy labels. For instance, one might use the loss correction approach:
> > > https://proceedings.neurips.cc/paper/2013/file/3871bd64012152bfb53fdf04b401193f-Paper.pdf
> > > to validate but then it is unclear how one should be setting the noise parameters for different approaches. Peer loss was one that promotes itself on validation too, which doesn't require the noise parameters:
> > > http://proceedings.mlr.press/v119/liu20e/liu20e.pdf.
> > > Some recent works proposed to stop at certain epochs to select models, but this gives unfair favor to models that explicitly require early stopping etc. Some might even require ground truth labels for the validation dataset.
> > >
> > > These differences pose challenges for choosing the right size of validation data, so to enable parameter estimates, making use of the required ground truth label for all methods in the right way when we compare, etc. Unfortunately, to our best knowledge, the community hasn't reached a consensus on which approach to go for when one wants to fairly compare models on a noisy validation set. Inconsistent comparisons were reported in the literature, as well as were observed in our own experiments too.  We think this is a highly important problem to address in the near future, but we will clarify the intention of our choice.

---

> ### Author Response · Authors · 2021-11-19
> **Response to Reviewer PASS (part3)**
>
> **Question 6:**  Somehow the authors mention the regularizer in section 4. but do not use it in Table 1. Why is that?
>
> **Response 6:**
> The purposes of Section 3 and Section 4 are different.
> Section 3 is mainly to show the benefits of only fine-tuning the linear classifier with **fixed SSL pre-trained encoder** and discuss the condition of achieving the consistent classifier. We provide theoretical and experimental results (Figure 3, Table 1) to support our claims.
> While our motivation of the regularizer in Section 4 is to consider the case where the **encoder can not be fixed** (See **Takeaways** in Section 3). In this case, due to the great over-fitting capability of DNN, directly using Cross Entropy is hard to achieve optimal classifiers, thus we introduce the structure regularization by utilizing SSL features.
>
> Experimentally, one can add the Regularizer when only fine-tuning linear classifier on noisy labels. However, due to the limited learnability of affine function, it is hard to learn the intrinsic structure of SSL features.
>
>
> **Question 7:** The paper can be re-organized. The authors describe the use of SSL first and then describe some experiments. After that, the authors describe the regularizer and some other experiments. They look like two separate parts. I suggest the authors put all the experiments and discussions in one section.
>
> **Response 7:** As answered in [A6], Section 3 shows the benefits of only fine-tuning the linear classifier with the fixed SSL pre-trained encoder while Section 4 discusses how to use SSL features when the encoder can not be fixed. **We summarized Section 3 and motivated Section 4 in Section 3.3 Takeaways.**
> We will better present our theories and experiments in the final version.

---

### Official Review · Reviewer_GRMb · 2021-11-02

**Correctness:** 3
**Technical Novelty And Significance:** 3
**Empirical Novelty And Significance:** 2
**Recommendation:** 5
**Confidence:** 4

**Main Review:**


Pros:
Providing a theoretical understanding of how self-supervised pre-trained features help to improve network resistance against noisy labels.
The paper conducted experiments that supported their theoretical claims

Cons:
The experiments were conducted under limited noisy ratio levels from 20 to 60 percent. No extreme noise levels were checked, i.e., >=80%. In addition, only symmetric synthetic noise scenario is consider, while more complicated asymmetric and real-life noise cases are not considered.


**Summary Of The Paper:**

This paper studies how self-supervised pre-training impacts the resistance of the neural network to noisy labels. The paper provides both theoretical analyses and numerical experiments of their study.  The main contribution of the provided study is (i) given a quality encoder pre-trained from SSL, a simple linear layer trained by the cross-entropy loss is theoretically robust to symmetric label noise (ii) providing insights for how knowledge distilled from SSL features can alleviate the over-fitting problem.

**Summary Of The Review:**

This paper aims to provide a theoretical understanding of why self-supervised pre-trained features help neural networks improve resistance under noisy labels. While the theoretical claims look solid, the important, more general settings outside of symmetric synthetic noise are missing (please see "Cons").
Hence, in the pre-rebuttal phase, my score is 5 and it can be re-evaluated once the authors will address my concerns

Post rebuttal decision:
I am grateful to the authors for their responses to the reviewers' concerns during the rebuttal phase. Nevertheless, the paper presents an important study the experimental setup is not comprehensive enough to support the paper's concussions, Hence, I vote to reject the paper in its current form and maintain my current score, I encourage the authors to expand the experimental study by my and other reviews comments.

---

> ### Author Response · Authors · 2021-11-19
> **Response to Reviewer GRMb**
>
> Thanks for reviewing our paper and providing suggestions. Please see our response below:
>
> **Question 1:** The experiments were conducted under limited noisy ratio levels from 20 to 60 percent. No extreme noise levels were checked, i.e., $>=80\%$.
>
> **Response 1:** Note for K-class classification with symmetric label noise, the noise ratio is bounded by $\frac{K-1}{K}$. In Figure 3(a), 3(b), and 3(c), we **had experiments on extreme noise levels** in our initial submission. For example, 0.45 for DogCat (binary classifications require noise rates $< 0.5$), and 0.8, 0.85 for CIFAR10 and CIFAR100. Note these experiments show empirical evidence that **a fixed encoder benefits extreme label noise** as detailed in Section 3.2.2. They also support our theoretical analyses in Section 3.2.1.
>
> In Figure 5, the noise level is from 20 to 60 percent. We have performed experiments with larger noise ratio (0.8 and 0.85) as follows:
>
>
>
> |       Method        | sym. 0.8 | sym. 0.85 |
> | :-----------------: | :------: | :-------: |
> |         CE          |   37.0   |   17.01   |
> | CE with Regularizer |  60.56   |   28.41   |
>
>
> It can be observed that the regularizer can also improve network robustness on extreme noise levels.
>
>
> **Question 2:**  Only symmetric synthetic noise scenario is considered, while more complicated asymmetric and real-life noise cases are not considered.
>
> **Response 2:**
>  First of all, we have implemented instance-dependent label noise (which is a more realistic and challenging setting than the symmetric and the asymmetric label noise) in Table 1 (in our initial submission). We also provide new theoretical and experimental results on these settings of label noise (See our response to Reviewer GhTK).
>
> For real-world label noise, we perform experiments on CIFAR100-N (A challenging human-annotated noisy dataset) [R1] to verify the effectiveness of structure regularization by using SSL features.
>
> | Method | CE    | Forawrd-T | GCE   | JoCoR | Peer Loss | ELR   | Structure Regularization |
> | ------ | ----- | :-------: | ----- | ----- | --------- | ----- | :----------------------: |
> | Accuracy    | 55.50 |   57.01   | 56.73 | 59.97 | 57.59     | 58.94 |          61.12           |
>
>
> It can be seen that by using SSL features, structure regularization can outperform many benchmark methods on the real-world human-annotated noisy dataset.
>
> [R1] http://noisylabels.com

---

### Official Review · Reviewer_GhTK · 2021-11-03

**Correctness:** 4
**Technical Novelty And Significance:** 3
**Empirical Novelty And Significance:** 3
**Recommendation:** 6
**Confidence:** 4

**Main Review:**

Strength: (1) Theoretical analysis on one kind of noisy-label learning method.
Weakness: (1) The empirical analysis could be improved (see below).

**Summary Of The Paper:**

The paper performs a theoretical analysis on one kind of noisy label learning method. The paper also presents an empirical analysis.


**Summary Of The Review:**

The experiment needs to be improved. The paper performs empirical analysis only on one setup. It could make the paper better if the authors could perform analysis on other settings (e.g., asymmetric noise, etc).

---

> ### Author Response · Authors · 2021-11-19
> **Response to Reviewer GhTK**
>
> Thanks for reviewing our paper and providing suggestions. Please see our response below:
>
>
> **Question 1:** The paper performs empirical analysis only on one setup. It could make the paper better if the authors could perform analysis on other settings (e.g., asymmetric noise, etc).
>
> **Response 1:**
>
> **Theoretically**, we extend Theorem 1 to multi-class classification in Section F to show that under symmetric label noise with equal noise ratio for each class, the classifier is consistent with the classifier trained on the clean dataset, *i.e.,* the decision boundaries are the same. We also provide theoretical proof in Section G to show that under asymmetric and instance-dependent label noise, the classifier is not consistent.
>
> **Experimentally**, we perform experiments on CIFAR-10 with asymmetric label noise to verify our theory as follows:
>
>
> |                     Method                      | asym. 0.1 | asym. 0.2 | asym .0.3 | asym. 0.4 |
> | :---------------------------------------------: | :-------: | :-------: | :-------: | :-------: |
> | CE (baseline, unfixed encoder with random init) |   90.69   |   88.59   |   86.14   |   80.11   |
> |       CE (fixed SSL pre-trained encoder)        |   90.85   |   89.92   |   88.25   |   82.46   |
>
>
> It can be seen that for asymmetric label noise, there are only marginal improvements which verify  our theoretical analysis that asymmetric label noise does not induce a consistent classifier. Our theory also explains the experimental results in [R1] that fine-tuning SSL features has more benefits on symmetric label noise than asymmetric label noise.
>
>
> For instance-dependent label noise (which is a more realistic setting than asymmetric label noise), we have shown in Table 1 (in our initial submission) that directly fine-tuning linear classifier does not bring benefits. However, a simple down-sampling strategy on instance-level noisy labels can approximate the condition in Theorem 1 and Theorem 5 which brings substantial improvements (Illustration can be found in Section C).
>
> We have added these new experiments, theorems and analyses in the revised version. We hope our analysis can facilitate a better understanding on dealing with label noise using SSL features.
>
>
> [R1] A. Ghosh and A. Lan. Contrastive learning improves model robustness under label noise. In Proceedings of the IEEE/CVF Conference on Computer Vision and Pattern Recognition, pages 2703–2708, 2021.

---

### Author Response · Authors · 2021-11-19
**General Response to All Reviewers**

We thank all reviewers' detailed comments and helpful suggestions. We have some updates for the paper to address some common concerns raised by the reviewers:


 - We extend Theorem 1 to multi-class classification in Section F and provide theoretical analysis on other types of label noise such as asymmetric and instance-dependent label noise in Section G.

- We perform experiments on the dataset with asymmetric and real-world human-annotated label noise to further verify our theorem and methods (Section D.7, Section G).

- **Takeaways** in Section 3 (in our initial submission) **explains the relationship between Section 3 and Section 4**. To make it more clear, we will add more motivations and explanations in our final submission.


Next, we respond to all the questions raised by each reviewer.  Please feel free to let us know if there is still any confusion.

---

### Decision · Program_Chairs · 2022-01-20

**Decision:**

Reject

**Comment:**

This paper presents an analysis of the robustness of self-supervised learning (SSL) features to noisy labels in downstream supervised learning, and provides empirical verification of the results (mostly in the symmetric noise setup); a SSL regularization scheme is also analyzed (section 4). While the paper contains plausible insights, the reviews share similar concerns that the analysis is mainly based on the noise being symmetric, and that the SSL features already have good class separation and Gaussian clusters, which are strong assumptions. Given that the assumptions are not theoretically verified, and that there is not sufficient empirical results in heavy non-symmetric noise scenario on large benchmark datasets, the reviewers think the paper does not provide practical guidance for noise label learning in its current form.